# Nuclear-capture of endosomes depletes nuclear G-actin to promote SRF/MRTF activation and cancer cell invasion

Sergi Marco [1], Matthew Neilson[1], Madeleine Moore[1], Arantxa Perez-Garcia[2], Holly Hall [1], Louise Mitchell[1], Sergio Lilla [1], Giovani R. Blanco [1], Ann Hedley [1], Sara Zanivan [1,2] & Jim C. Norman [1,2 ✉]

Signals are relayed from receptor tyrosine kinases (RTKs) at the cell surface to effector systems in the cytoplasm and nucleus, and coordination of this process is important for the execution of migratory phenotypes, such as cell scattering and invasion. The endosomal system influences how RTK signalling is coded, but the ways in which it transmits these signals to the nucleus to influence gene expression are not yet clear. Here we show that hepatocyte growth factor, an activator of MET (an RTK), promotes Rab17- and clathrin-dependent endocytosis of EphA2, another RTK, followed by centripetal transport of EphA2-positive endosomes. EphA2 then mediates physical capture of endosomes on the outer surface of the nucleus; a process involving interaction between the nuclear import machinery and a nuclear localisation sequence in EphA2's cytodomain. Nuclear capture of EphA2 promotes RhoG-dependent phosphorylation of the actin-binding protein, cofilin to oppose nuclear import of G-actin. The resulting depletion of nuclear G-actin drives transcription of Myocardin-related transcription factor (MRTF)/serum-response factor (SRF)-target genes to implement cell scattering and the invasive behaviour of cancer cells.

[1] CRUK Beatson Institute, Glasgow G61 1BD Scotland, UK. [2] Institute of Cancer Sciences, University of Glasgow, Glasgow G61 1QH Scotland, UK. ✉email: j.norman@beatson.gla.ac.uk

Many plasma membrane receptors, including receptor tyrosine kinases (RTKs), are endocytosed following ligand engagement and are trafficked through the endosomal system in a way which is thought to influence signalling outcomes[1]. It is established that signal termination can occur after endocytosis, where ubiquitinated receptors are sorted for lysosomal degradation[2]. More recently, an increasingly complex picture is emerging in which certain endosomal compartments and membrane subdomains, including late and recycling endosomes, constitute platforms that influence downstream signalling coding[3]. Many of these endosomal compartments are positioned very close to the nucleus and this may facilitate communication of signals to the transcriptional machinery. An early study demonstrated that the Rab5 effector, APPL, translocates to the nucleus, wherein it interacts with the nucleosome remodelling machinery to drive transcription and cell proliferation. In this case, internalization of the RTK, epidermal growth factor receptor 1 (EGFR1) releases APPL from Rab5 and, by doing so, APPL is free to shuttle into the nucleus to interact with the histone deacetylase complex NuRD/MeCP1 to regulate chromatin structure and gene expression[4,5]. The involvement of endosomal trafficking in positioning a signalling event near the nucleus was subsequently described for mesenchymal-epithelial transition factor (MET). MET is an RTK for hepatocyte growth factor (HGF), which plays key roles in transducing signals that lead to increased cancer cell invasion and metastasis, and much signalling downstream of MET is thought to be mediated via activation of the oncogene activator of transcription, STAT3. Microtubule-dependent transport moves endosomes containing MET into close proximity of the nucleus to co-localize with the STAT3 transcription factor[6]. The close juxtaposition of this complex to the nucleus has been proposed to minimize signal dissipation, thus facilitating efficient activation of the transcriptional machinery in the nucleus.

There are numerous reports of RTKs, and the intracellular domains of RTKs, being imported into the nucleus to perform signalling roles[7]. RTKs such as fibroblast growth factor receptor 1 or platelet-derived growth factor receptor-β have been shown to be translocated as holoreceptors into the nucleus, whereas apparently intact ErbB2 (Erb-B2 RTK 2), VEGFR1 (vascular endothelial growth factor receptor 1) or MET have not only been shown to be found in the nuclear milieu, but also their C-terminal portions may be delivered to the nucleus following cleavage in the cytosol by enzymes such as secretases[8]. RTKs have been shown to perform several roles in the nucleus, with transcriptional coregulation being prominent among the processes thought to be downstream of nuclear-imported RTKs and their C-terminal portions. Nevertheless, the nuclear translocation of transmembrane receptors poses several topological 'questions'—such as 'how do transmembrane receptors cross the nuclear membranes?' and, perhaps more challengingly, 'what is the membrane domain topology of a type I membrane protein when it resides within the nucleoplasm?'. Mechanistic information enabling us to address these questions had been scarce until, more recently, EGFR1 was shown to be delivered to the nucleoplasm via docking and fusion of endosomes with the nuclear membrane[9,10]. This study identified a pathway that transports internalized EGFR1 through the early endosomal system towards the nucleus, whereupon EGFR1-containing endosomes associate physically with the outer nuclear membrane. These nuclear-associated endosomes then discharge their content into the nucleoplasm in a manner dependent upon the SUN1/SUN2 nuclear envelope proteins. Interestingly, these investigators found that the SUN/SUN2 complex delivered EGFR1 itself to the nucleus. Thus, although questions remain as to the topology of the nuclear RTK, this study has provided substantive evidence for a mechanism through which the endosomal system may directly interface with gene expression.

This body of work prompted us to further investigate endosomal pathways that might physically link external cues at the plasma membrane level to transcriptional responses in the nucleus. In this work we develop a quantitative proteomic method to pin-point key endosomal cargoes, which are translocated from the plasma membrane to the nucleus following activation of MET by the addition of HGF. This indicates that HGF promotes endocytosis of EphA2, another RTK, followed by capture of EphA2-positive endosomes upon the nuclear surface, in an interaction mediated by a nuclear localization signal (NLS) in the cytoplasmic domain of EphA2. This event promotes both actin polymerization in the juxta-nuclear region and phosphorylation of the actin monomer-binding protein, cofilin. We then proceed to use a combination of cell biological and mathematical modelling approaches, to determine that it is primarily the ability of nuclear-captured EphA2 to phosphorylate cofilin leading to depletion of nuclear G-actin, which drives changes in gene expression to enable cell migration and invasion.

## Results

**EphA2 mediates capture of endosomes at the nuclear surface.** To study the influence of MET signalling on the intracellular destination of plasma membrane cargoes, we surface-labelled H1299 cells with a cell-impermeant biotinylation reagent at 4 °C. We rapidly warmed the cells to 37 °C, to allow internalization of cell surface proteins, and this was conducted in the presence and absence of HGF. Fluorescence microscopy was then used to image the distribution of internalized biotinylated material. This revealed a population of endosomes transporting plasma membrane-derived material into very close proximity of the nuclear membrane and this population was significantly increased by HGF addition (Fig. 1a).

These observations prompted us to develop a stable isotope labelling (SILAC)-based mass spectrometry (MS) proteomic approach to identify the cargoes trafficking from the plasma membrane to endosomes that are physically associated with the nucleus. H1299 cells were SILAC-labelled with medium (M) or light (L) amino acids and surface-biotinylated at 4 °C. Internalization of cell surface proteins was then triggered by increasing the temperature to 37 °C in the presence or absence of HGF for 30 min. Following this, nuclei and their associated endosomes were purified, biotinylated proteins isolated from these using streptavidin beads and the proteome of the isolates analysed by MS (Fig. 1b diagram). Western blotting (WB) confirmed that the purified nuclear preparations were uncontaminated with membranes from the endoplasmic reticulum and Golgi, and from cytoplasmic components (Fig. 1c). The results obtained indicated that activation of MET by HGF promoted translocation of a number of proteins, mainly receptors, from the plasma membrane to the nucleus, and another RTK, EphA2, was among these (Fig. 1b and Supplementary Data 1). To confirm this, we developed an approach to quantify the delivery of surface receptors to the nuclear fraction. We biotinylated cell surface proteins at 4 °C and warmed the cells to 37 °C in the presence and absence of HGF, as for Fig. 1a, to allow internalization of surface receptors. Following cooling back to 4 °C, we then purified nuclei (as for Fig. 1b, c) and used a capture-enzyme-linked immunosorbent assay (ELISA) to quantify the biotinylated-EphA2, which had relocated from the cell surface to this subcellular fraction. This indicated that ~5% of surface EphA2 moved from the cell surface to the nucleus within 15 min for internalization, and that this was significantly increased in the presence of HGF (Fig. 1d).

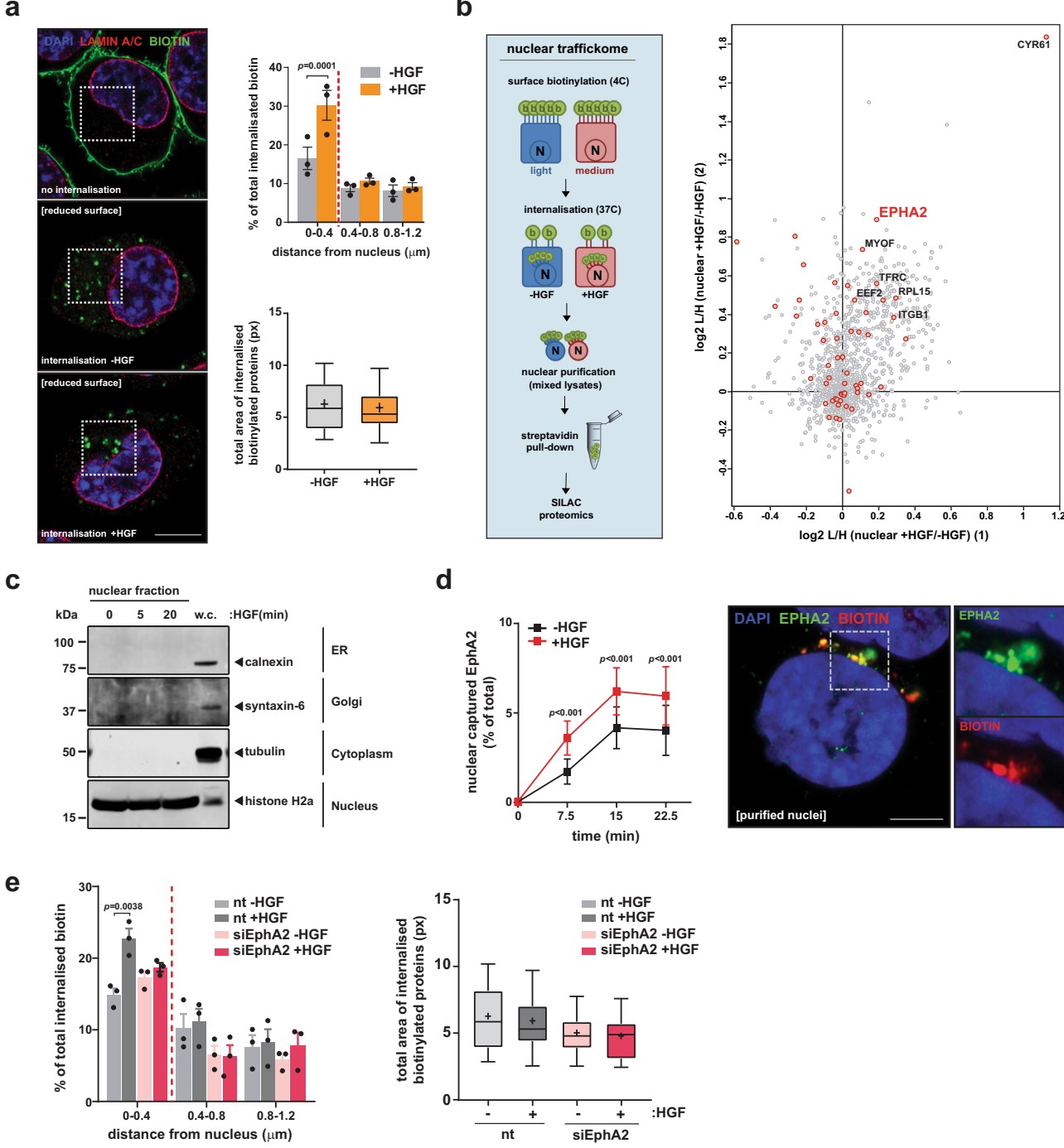

Super-resolution Airyscan microscopy confirmed that this nuclear-associated EphA2 was present in vesicles that were attached to the nuclear surface (Fig. 1d). Furthermore, small interfering RNA (siRNA) knockdown of *EPHA2* opposed recruitment of plasma membrane-derived vesicles to the nuclear surface, indicating that EphA2 is not a passive cargo of this pathway, but that it is necessary for their delivery to and/or attachment of these endosomes to the nucleus (Fig. 1e).

Ephrin ligand-independent functions of EphA2 are known to be mediated by phosphorylation of $Ser^{897}$ in its cytotail[11,12]. $Ser^{897}$ is phosphorylated following activation of signalling downstream of several growth factor including epidermal growth factor, basic fibroblast growth factor, platelet-derived growth factor, tumour necrosis factor-α (TNFα) and HGF[11]. Moreover,

these authors demonstrate that this is owing to the ability of these growth factors to drive signalling pathways leading to activation of Akt, although TNFα's capacity to drive increased phospho-$Ser^{897}$ more likely proceeds via activation of ribosomal S6 kinase (and not Akt)[12]. HGF is long-established to drive Akt signalling via engagement of MET[13] and we have previously shown that HGF-driven internalization of EphA2 requires Akt-driven phosphorylation of EphA2 at $Ser^{897}$ in its cytotail[14]. To further elucidate factors required for HGF-driven packaging of EphA2 into endosomes, we deployed an siRNA to oppose clathrin heavy chain expression (Fig. 2a) and which blocks endocytosis of the transferrin receptor (the hallmark cargo of clathrin-coated pits) (Fig. 2b). This indicated that EphA2 internalization (both in the absence and presence of HGF) is strongly clathrin-dependent

**Fig. 1 HGF promotes nuclear-capture of EphA2-positive endosomes. a** H1299 cells were surface-biotinylated and allowed to internalize in the presence or absence of HGF. Biotin remaining at the cell surface was removed and distribution of biotinylated proteins visualized with streptavidin (green) and counterstained with DAPI (blue) and LaminA/C (red). Scale bar, 10 μm. Internalized particles (lower graph) and their distance from the nuclear surface (upper graph) was quantified. Values are mean ± SEM (upper graph) ($n = 3$ individual experiments), statistical significance was determined by one-way analysis of variance (ANOVA). Box and whiskers: 10–90 percentile whiskers, + represents mean, black line represents median (lower graph). **b, c** SILAC-labelled cells were surface-biotinylated and internalized as for **a**. Nuclei were purified and the biotinylated proteome determined by mass spectrometry (schematic). Western blotting confirmed integrity of the nuclear preparations (**c**). The scatter plot indicates the SILAC ratios of the biotinylated nuclear-associated proteome (+HGF/−HGF) from two independent experiments ((1) and (2)) plotted on the x and y-axes, respectively. Biotinylated proteins enriched in the nuclear fraction in response to HGF are represented in the upper right-hand quadrant. Proteins moving from the plasma membrane to the nuclear fraction are highlighted by red dots. **d** Surface-biotinylated cells were allowed to internalize in the absence or presence of HGF for the indicated times. Nuclei were purified and the presence of biotinylated-EphA2 in these purified nuclei determined using a capture-ELISA (left graph) and by super-resolution microscopy (right panels, scale bar 5 μm). Data are expressed as the proportion of surface-labelled EphA2, which is translocated to the nuclear preparation. Values are mean ± sem, $n = 3$ individual experiments, statistical test is repeated-measures two-way ANOVA. **e** H1299 cells were transfected with siRNAs targeting *EPHA2* (siEphA2) or a control (nt), surface-labelled and allowed to internalize in the presence or absence of HGF. Internalized biotinylated particles were determined as for **a**. Values are mean ± SEM (left graph) ($n = 3$ individual experiments), statistical significance was determined by two-way analysis of variance (ANOVA). Box and whiskers: 10–90 percentile whiskers, + represents mean, black line represents median (right graph).

(Fig. 2c). We next tested the involvement of a battery of Rab GTPases and other endosomal regulators known to be involved in cell adhesion and migration (including Rabs 4, 5, 7, 11, 17, 21 and 25, CD63 and LAMP1/2). This highlighted Rab17 as being required for both HGF-driven internalization (Fig. 2d) and nuclear-capture (Fig. 2e) of EphA2. Super-resolution imaging also enabled us to identify vesicles containing both EphA2 and Rab17 at the membrane of purified nuclei (Fig. 2f). Moreover, live-cell fluorescence imaging indicated that Rab17 accompanied EphA2 on its centripetal journey from the plasma membrane to vesicles captured at the nuclear surface (Fig. 2g and Supplementary Movie 1).

**EphA2 contains a juxtamembrane NLS required for nuclear-capture of endosomes.** NLS are peptide sequences found in proteins that translocate from the cytoplasm to the nucleus. NLSs are recognized by importin-α, which in turn recruits importin-β nuclear receptors to promote the association of this complex with the nuclear pore to drive nuclear import[15]. As a bioinformatic analysis[16] predicted a putative NLS in the juxtamembrane region of EphA2, we generated two mutants of this receptor (NLS1 and NLS2), designed to disrupt key basic residues responsible for importin-α recognition (Fig. 3a). As a proof-of-principle, we first determined that fusion of EphA2's putative NLS with green fluorescent protein (GFP) (NLS-GFP WT), increased nuclear delivery of GFP (Supplementary Fig. 1a). Furthermore, EphA2-GFP fusion proteins in which the NLS was mutated (NLS-GFP[NLS1] and NLS-GFP[NLS2]) did not concentrate in the nucleus (Supplementary Fig. 1a), demonstrating that EphA2's NLS-like sequence is likely to mediate functional interaction with the nuclear import machinery.

To further explore how EphA2 might interact with the nuclear import machinery, we deployed a proximity-labelling approach (Fig. 3b). We generated constructs in which either wild-type or NLS-mutated EphA2 was fused to an engineered biotin ligase, TurboID, which catalyses addition of biotin to proteins within its close proximity (<10 nm). These biotinylated proteins are then recovered using streptavidin beads to yield the proximal proteome of EphA2 (Fig. 3b). We confirmed that, following stable expression in H1299 cells, these TurboID-EphA2 fusion proteins appropriately localized to both the plasma membrane and endosomes (Supplementary Fig. 1b). Moreover, immuno-blotting indicated that both the wild-type and NLS-mutated TurboID-EphA2 fusion proteins were efficiently expressed, and that they were able to efficiently autobiotinylate (Supplementary Fig. 1c). H1299 cells expressing wild-type or NLS mutant EphA2-TurboID constructs were then incubated with biotin in the

presence of HGF and the resulting biotinylated proximal proteome was then determined using MS-based proteomics. This analysis indicated that the proximity proteome of EphA2 comprised several nuclear pore complex and nuclear pore-related proteins, and that interaction of these with EphA2 was favoured when its NLS was intact (Fig. 3c, Supplementary Fig. 1d and Supplementary Data 2). We validated a selection of these nuclear pore hits by WB and this demonstrated that EphA2 interacted with proteins located at the outer and central pore and cytoplasmic ring of the nuclear pore complex, but not with an abundant underlying inner nuclear membrane protein, such as laminA/C (Fig. 3d, e).

As it is the case for many NLS-containing cytoplasmic proteins that are recruited to the nuclear pore, EphA2 co-immunoprecipitated with both α- and β-importin subunits (including α5, β1 (Fig. 3f) and α7 importins and transportin-3 (Supplementary Fig. 1e) heterodimers) after HGF addition in a manner which was opposed by mutation of its NLS sequences (Fig. 3f and Supplementary Fig. 1e). Consistently, fluorescence time-lapse microscopy indicated that EphA2-positive vesicles appeared to contact structures that were positive for importin-α5 and β1-positive structures located near the nuclear surface (Fig. 3g and Supplementary Movies 2 and 3). Importantly, surface biotinylation/capture-ELISA approaches indicated that mutation of EphA2's NLS opposed nuclear-capture of EphA2 vesicles (Fig. 3h), without compromising canonical signalling downstream of EphA2 (Supplementary Fig. 1f). Taken together, these data indicate that addition of HGF promotes clathrin- and Rab17-dependent endocytosis of EphA2. Following this, EphA2 and Rab17-positive endosomes are transported centripetally to associate with the nuclear pore in an interaction that requires a functional NLS in the cytoplasmic tail of EphA2.

**Nuclear-capture of endosomes is required for HGF-driven cell scattering and cancer cell invasion.** Evidence is accumulating for a key role for EphA2 in the acquisition of invasive and metastatic behaviour in cancer[14,17–20]. Indeed, we have previously reported that an autochthonous model of pancreatic ductal adenocarcinoma (PDAC)—the 'KPC' model, which is driven by expression of mutants of *Kras* (K) and *Trp53* (P) under control of a pancreatic-specific *Cre* (C) recombinase—display reduced metastasis when conducted in EphA2-knockout mice[14]. Consistently, cells cultured from EphA2-knockout KPC tumours (KPC-PDAC cells) display profoundly reduced ability to invade into Matrigel towards a gradient of HGF (Fig. 4a). Invasiveness of cells from EphA2-knockout KPC cells was restored by expression of wild type, but not NLS mutants of EphA2 (Fig. 4a), indicating

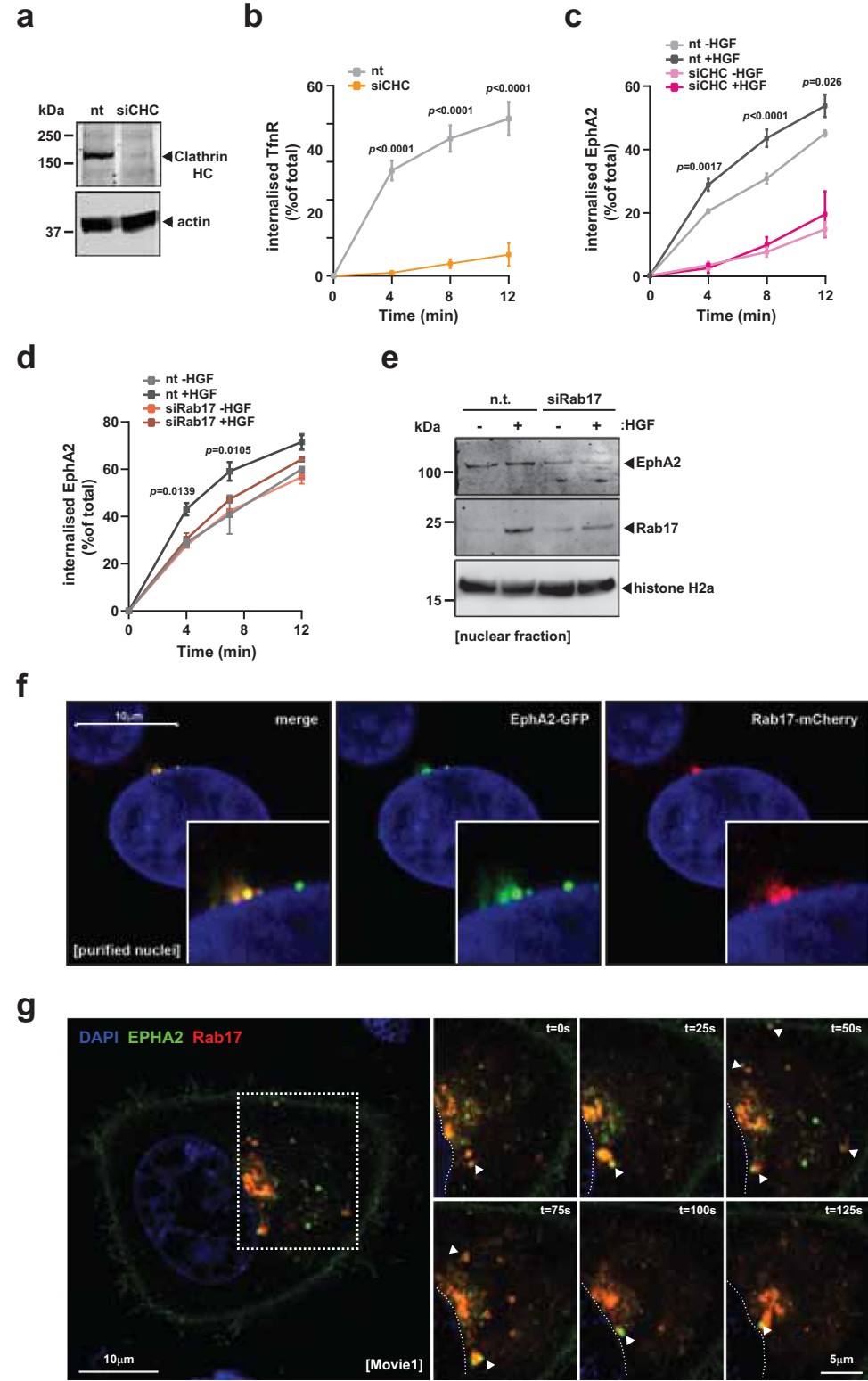

that nuclear-capture of endosomes contributes to an HGF-driven migratory process. Accordingly, cell scattering—the canonical cellular response to activation of HGF/MET signalling—was opposed by knockout (Fig. 4b, c) or siRNA (Fig. 4d–f) of *EPHA2* in KPC-PDAC and H1299 cells, respectively, and this was rescued by expression of wild-type, EphA2, but not its NLS mutants. Moreover, siRNA of *RAB17* (to reduce internalization of EphA2) opposed HGF-driven cell scattering (Fig. 4f). Taken together,

these data indicate that internalization, centripetal transport, and NLS-mediated nuclear-capture of EphA2-positive endosomes is required for cells to mount a cell scattering and invasive response following addition of HGF.

**EphA2-mediated nuclear-capture influences expression of MRTF/SRF target genes.** Engagement of MET with HGF exerts post-transcriptional influence over the cell's migratory

**Fig. 2 Clathrin and Rab17 control internalization and intracellular trafficking of EphA2. a–d** H1299 cells were transfected with siRNA targeting clathrin heavy chain (siCHC), Rab17 (siRab17) or control (nt). Surface-biotinylated cells were allowed to internalize for the indicated times in the absence or presence of HGF and biotin remaining at the cell surface was removed. Internalized, biotinylated transferrin receptor (TfnR) (**b**) or EphA2 (**c, d**) were determined using capture-ELISA. Values are mean ± SEM, $n = 3$ independent experiments, statistical test is two-way analysis of variance (ANOVA). Knockdown of clathrin was determined by western blotting with actin as a loading control (**a**) and knockdown of Rab17 using qPCR (see Supplementary Fig. 3d). **e** Rab17 (siRab17) or control (nt) knockdown cells were incubated in the presence or absence of HGF for 5 min. Nuclei were purified, and EphA2 and Rab17 in these purified nuclei were determined using western blotting with histone H2a as loading control. **f** Cells expressing EphA2-GFP and Rab17-mCherry were incubated with HGF for 5 min. Nuclei were purified and the distribution of GFP (green) and mCherry (red) visualized using super-resolution microscopy; scale bar, 10 μm. **g** Cells expressing EphA2-GFP (green) and Rab17-mCherry (red) were challenged with HGF and movies collected using live confocal time-lapse fluorescence imaging (Supplementary Movie 1). Scale bar, 10 μm for the main panel and 5 μm for the insets. Stills (captured at the indicated times following HGF addition) from these movies are presented in the insets. The arrowheads indicate EphA2/Rab17-positive endosomes budding from the plasma membrane or moving towards the perinuclear region.

machinery—including actin polymerization and the turnover of cell-matrix adhesions[21]. In addition, HGF signalling has an established role in driving transcriptional programmes, which command invasive growth and metastatic behaviour[22]. Our finding that nuclear-capture of EphA2-positive endosomes is mediated via direct physical interaction with the nuclear pore, prompted us to consider the possibility that this process may be involved in transducing signals into the nucleus to drive a pro-invasive transcriptional gene expression programme. To establish whether transcriptional activity is required for invasive responses, we treated cells with actinomycin D (to inhibit RNA synthesis) and measured cell scattering following the addition of HGF. This clearly indicated that, although cells do retain the ability to migrate following inhibition of transcription (not shown), scattering of H1299 cells in response to HGF is completely opposed by the addition of actinomycin D (Supplementary Fig. 2). We, therefore, used RNA sequencing (RNAseq) to profile gene expression signatures driven by HGF and to determine whether components of this are dependent on EphA2 (Supplementary Data 3). This indicated that several HGF-responsive mRNAs were sensitive to siRNA of *EPHA2* (Fig. 5a left panel). Serum-response factor (SRF) and myocardin-related transcription factor (MRTF) targets[23,24] were strongly represented in this cohort of genes (Fig. 5a, right panel—highlighted in red). Four of the SRF targets highlighted by RNAseq (*ATF3*, *FOSB*, *JUNB*, and *ZFP36*) were further validated by qPCR, confirming that in both H1299 (Fig. 5b) and KPC-PDAC cells (Fig. 5c), HGF-driven expression of these genes was restored by re-expression of wild-type EphA2, but not when the cells expressed its NLS mutant counterparts.

Similarly, knockdown of clathrin heavy chain, Rab17 and the cytoplasmic filament-associated component of the nuclear pore, RanBP2, opposed HGF-driven increases in *ATF3*, *FOSB*, *JUNB* and *ZFP36* expression (Supplementary Fig. 3a–e), consistent with a role for endocytosis, centripetal transport, and nuclear-capture of EphA2-positive endosomes in the ability of MET to communicate with the gene expression machinery. In addition, HGF-driven expression of other well-established MRTF targets, including SRF itself and several cytoskeletal genes[23], was opposed by siRNA of *EPHA2* (Supplementary Fig. 3f), corroborating the influence of EphA2 and nuclear-capture on activation of broad range of MRTF target genes. To further interrogate this, we performed a series of chromatin immunoprecipitation[24] assays that showed increased binding of MRTF (but not Elk-1) to *FOSB* and *JUNB* gene promoters following HGF addition, and that this was opposed by *EPHA2* knockdown (Fig. 5d and Supplementary Fig. 3g). Although this clearly demonstrates a requirement for EphA2 signalling in HGF-driven recruitment of MRTF to its target genes, it remains unclear how *EPHA2* knockdown might impact on MRTF/SRF protein levels under basal conditions. Finally, HGF-driven cell scattering was significantly inhibited by

siRNA knockdown of *SRF*, indicating that regulation of gene expression links nuclear-capture to HGF-driven cellular responses (Fig. 5e and Supplementary Fig. 3h).

**Nuclear-capture influences MRTF/SRF target gene expression by depleting nuclear G-actin.** MRTF is inhibited by association with monomeric actin (G-actin) and when actin polymerization is upregulated this activates gene expression by reducing the pool of G-actin available to inhibit MRTF[25]. We, therefore, considered the possibility that nuclear-capture may influence MRTF/SRF-dependent transcription by promoting actin polymerization. Live-cell imaging indicated that HGF drove actin polymerization in the vicinity of nuclear-captured EphA2 (Fig. 6a top panel and Supplementary Movie 4) and this was opposed by the expression of an NLS mutant of EphA2 (Fig. 6a left bottom panel and Supplementary Movie 5). EphA2 is known to promote actin polymerization and one mechanism through which it achieves this is via recruitment of ephexin4—a guanine nucleotide exchange factor (GEF) for the small GTPase, RhoG[26]. Activated RhoG, in turn, recruits ELMO2 and Dock4—a GEF for Rac1[27]—to drive actin polymerization. We confirmed the association between RhoG and EphA2 by demonstrating that these two proteins coimmunoprecipitate and additionally found that RhoG-EphA2 coimmunoprecipitation was enhanced by the addition of HGF (Supplementary Fig. 4a). We then used siRNA to suppress levels of RhoG and found that this reduced levels of F-actin in the juxta-nuclear region measured in the presence of HGF (Fig. 6a right bottom panel and Supplementary Movie 6). Consistently, siRNA targeting RhoG (*ARHGEF16*) opposed both transcription of SRF/MRTF target genes (Supplementary Fig. 4b, d) and HGF-driven cell scattering (Supplementary Fig. 4c), in the same way as did suppression of EphA2.

Given that MRTF activation is driven by the reduced monomeric actin levels that accompany actin polymerization, we used DNaseI (which, at low concentrations, binds specifically to monomeric actin) to visualize the distribution of G-actin in H1299 cells[28]. This indicated that, following addition of HGF, G-actin levels appeared to be reduced in the nucleus, but not in the surrounding cytoplasm (Fig. 6b). We next developed an approach in which we purified nuclei from cells that had previously been incubated in the presence or absence of HGF and used DNaseI staining followed by flow cytometry to quantify the HGF-driven changes to nuclear G-actin. This confirmed that addition of HGF significantly decreased nuclear G-actin levels (Fig. 6c left panel). Importantly, this was opposed by siRNA of *EPHA2* and rescued by wild type, but not an NLS mutant of, EphA2 (Fig. 6c right panel). Taken together, these data indicate that nuclear-capture of endosomes allows HGF signalling to promote actin polymerization in the juxta-nuclear cytosol and to drive depletion of nuclear (but not cytosolic) G-actin.

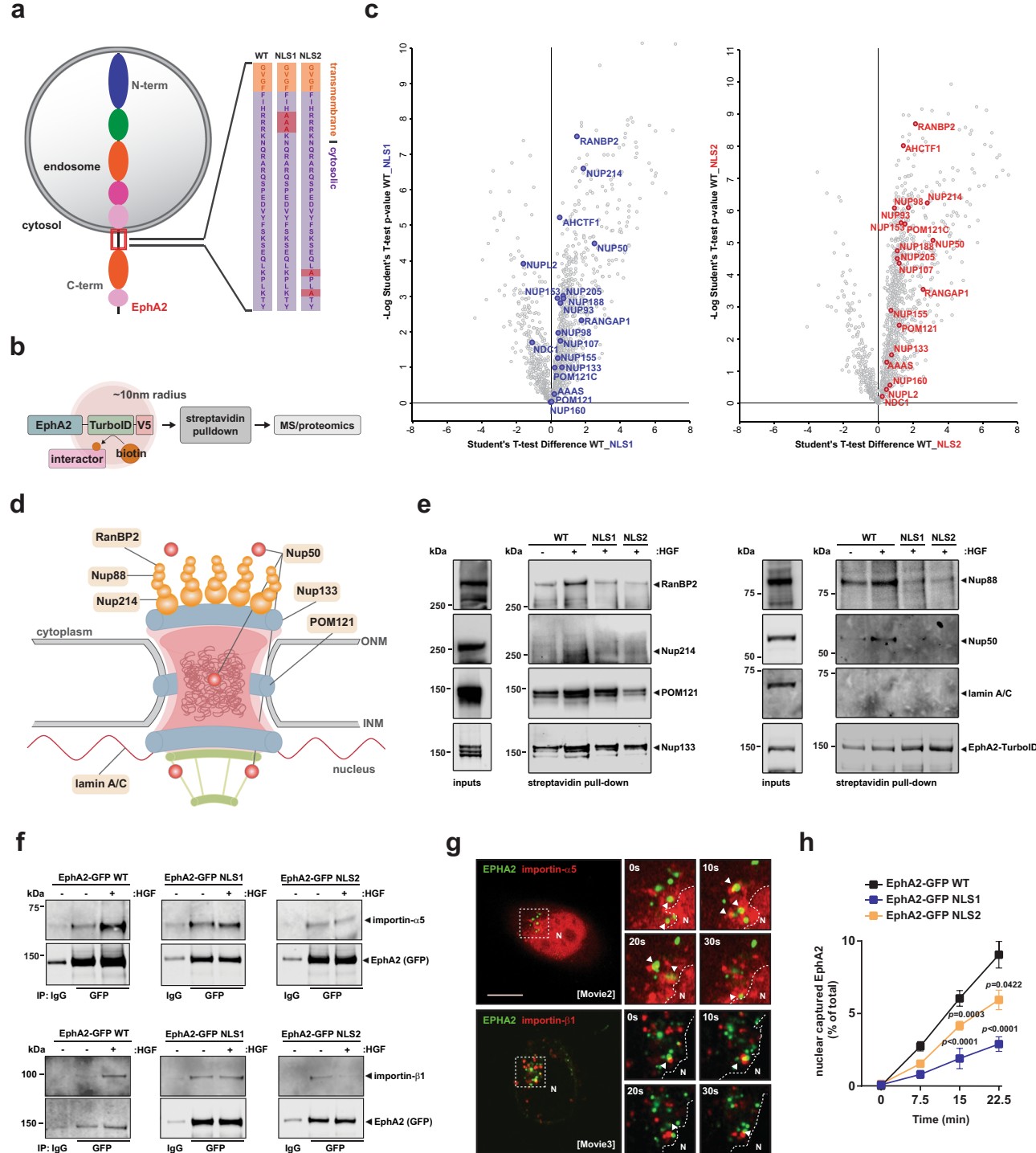

G-actin influences both nucleocytoplasmic shuttling[29] and transcriptional activity[30] of MRTF/SRF, so we used high-content imaging to measure nuclear shuttling of MRTF and decipher the role that EphA2, RhoG and G-actin might play in this. As expected, MRTF translocated to the nucleus following HGF addition. However, this was not opposed by knockdown of either EphA2 or RhoG (Fig. 6d left and centre panels). Moreover, by deploying non-polymerisable monomeric actins—actin$^{R62D}$ and the nuclear-targeted, NLS-actin$^{R62D}$ actin[31]—we found that nuclear shuttling of MRTF was unaffected by levels of actin monomer in the cytoplasm and nucleus, respectively (Fig. 6d

right panel). Despite this, the expression of actin$^{R62D}$ profoundly inhibited MRTF target gene expression (Supplementary Fig. 4e) and HGF-driven cell scattering (Fig. 6e). These data indicate that, although G-actin levels do not regulate nuclear translocation of MRTF, it is depletion of nuclear G-actin downstream of nuclear-capture that triggers MRTF transcriptional activity in response to HGF. Consistently, when we elevated nuclear G-actin levels by knocking down the exportin-6 transporter (XPO6) (Supplementary Fig. 4f, g), which mediates efflux of G-actin from the nucleus[32], HGF-driven transcription of MRTF target genes and cell scattering were inhibited (Supplementary Fig. 4h, i).

**Fig. 3 EphA2 interacts with the nuclear import machinery via a nuclear localization sequence in its cytotail. a** Schematic diagram of the sequence of endosomally localized EphA2, indicating its predicted juxtamembrane NLS sequence and amino acid substitutions deployed to generate NLS1 and NLS2 mutants. **b** Schematic diagram of the EphA2-TurboID construct and workflow to determine the proximity interactome of EphA2. **c** H1299 cells stably expressing EphA2-TurboID constructs with wild-type EphA2 (WT) or mutated EphA2 NLSs (NLS1 and NLS2) were incubated with biotin for 1 h. Biotinylated proteins were isolated and analysed using MS-based proteomics (**b**). Scatter plots display the biotin-labelled protein enrichment (*x*-axis) commanded by wild-type EphA2 by comparison with EphA2s harbouring mutated NLS sequences (NLS1 and NLS2). Significance scores (−Log Student's *t*-test *p*-value, *n* = 5 independent experiments) are plotted on the *y*-axes. Student's *t*-test values > −Log 1.4 (<*p* = 0.05) are significant. Proteins annotated correspond to nuclear pore components that were identified in the proximity interactome. **d** Schematic diagram of the nuclear pore complex. Annotated proteins correspond to EphA2 proximity interactors confirmed in **e**. **e** Cells expressing EphA2-TurboID constructs were incubated with biotin in the presence or absence of HGF for 1 h. Biotinylated target proteins were analysed using western blotting. **f** Cells expressing the indicated GFP-tagged EphA constructs were incubated in the presence or absence of HGF for 5 min. EphA2-GFPs were immunoprecipitated and the presence of importins -α5 and -β1 was determined using western blotting. Blots are representative of four independent experiments. **g** Cells expressing EphA2-GFP (green) in combination with mCherry-importin-α5 (upper panels, red) or mCherry-importin-β1 (lower panels, red) were imaged by fluorescence time-lapse microscopy and movies collected (Supplementary Movies 2 and 3). Stills from these movies (selected at 10 s intervals) are displayed in the right panels. Scale bar, 10 μm. **h** Cells expressing the indicated GFP-tagged EphA2 constructs were treated with HGF for the indicated times and delivery of the tagged EphA2s to the nuclear fraction was determined using capture-ELISA as for Fig. 1d. Values are mean ± SEM, *n* = 5 independent experiments, statistical test is repeated-measures two-way ANOVA.

**Nuclear-capture promotes depletion of nuclear G-actin via phosphorylation of cofilin.** Given that nuclear G-actin levels appear to be the lynchpin event in transducing signals between engagement of MET and activation of MRTF/SRF, we needed to define the role of EphA2 and its capture in depleting nuclear G-actin following HGF addition. To approach this, we considered several reactions that are known to influence nucleocytoplasmic shuttling of actin and these are summarized schematically in Fig. 7a. First, the rate of juxta-nuclear actin polymerization (denoted by $E_c\mu1$ in Fig. 7a), which we have found to be nuclear-capture and RhoG-dependent, is likely to affect availability of G-actin for nuclear import. Indeed, a recently described pathway linking mechanical force to gene expression in the skin invokes tension-induced F-actin polymerization in the immediate vicinity of the nuclear membrane, which restricts availability of G-actin in the nucleus to oppose transcription in a chromatin-dependent manner[33], and such a mechanism may also influence SRF/MRTF target gene expression. Another event that might influence nucleocytoplasmic actin dynamics is phosphorylation of cofilin at Ser[3] (by LIM domain kinase (LIMK)) (denoted by $E_c\mu2$ in Fig. 7a)[34], as this opposes formation of the cofilin–actin complex, which is the species transported into the nucleus via importin-9[35]. Indeed, addition of HGF increased cellular levels of phospho-cofilin and this was opposed by siRNA of EphA2 or RhoG and rescued by expression of wild-type (but not NLS mutants of) EphA2 (Fig. 7b). This indicates that cofilin phosphorylation is controlled by nuclear-capture and thus may influence communication with the transcriptional machinery.

We next used the reactions depicted in Fig. 7a to design a computational approach for modelling nucleocytoplasmic actin dynamics, and thus test the contribution of signalling events occurring downstream of EphA2 and RhoG to the depletion of nuclear G-actin. We modelled the cell as two static circular compartments representing the cytoplasm ($\Gamma_c$) and the nucleus ($\Gamma_n$) (Supplementary Fig. 5a). Then, for the species indicated in Fig. 7a (G-actin, F-actin, profilin, profilin–actin, cofilin, cofilin–actin), we assigned values of relative initial concentrations in both the nuclear and cytoplasmic compartments in a way that was consistent with experimental observations. We also generated equations representing the rates of exchange of these species across the nuclear membrane (as informed by published observations) and diffusion rates of these species within the nuclear and cytoplasmic compartments. Finally, we generated equations for the reactions indicated in Fig. 7a, which catalyse the interactions between, and the interconversion of, the various

species. All these designations and equations representing diffusion, intercompartmental transport and species interconversion are defined in the 'Methods'. We then allowed this computational model to run and present the results using the Viridis colour palette, which is used to display the relative concentrations of G-actin (Fig. 7c and Supplementary Movie 7). This model clearly predicted that when juxta-nuclear actin polymerization ($E_c\mu1$) and cofilin phosphorylation ($E_c\mu2$) (Fig. 7a) are both active ($E_c\mu1$ and $E_c\mu2 > 0$), this drives rapid depletion of nuclear G-actin and modestly increased cytosolic G-actin (Fig. 7c and Supplementary Movie 7), thus recapitulating the actin dynamics observed experimentally following HGF addition (as in Fig. 6b, c). We then tested the consequences of independently reducing either $E_c\mu1$ or $E_c\mu2$ to zero—i.e., mimicking selective inhibition of cytosolic actin polymerization ($E_c\mu1 = 0$) and/or inactivating cofilin phosphorylation ($E_c\mu2 = 0$), respectively. This indicated that cofilin phosphorylation ($E_c\mu2$) was likely to exert a greater influence over the depletion of nuclear G-actin depletion than actin polymerization ($E_c\mu1$) (Fig. 7c and Supplementary Fig. 5b). This prediction was then confirmed experimentally, as treatment with the Arp2/3 inhibitor, CK-666[36], completely opposed juxta-nuclear actin polymerization without opposing the expression of MRTF target genes (Supplementary Fig. 5c, d). Conversely, addition of the LIMK inhibitor, BMS-5[37], to oppose cofilin phosphorylation, completely opposed the ability of HGF to drive both expression of MRTF target genes and cell scattering (Fig. 7d, e). Taken together, these computational and experimental biology approaches indicate that HGF-driven nuclear-capture of EphA2 facilitates transmission of signals to the transcriptional machinery in the nucleus primarily by enabling phosphorylation of cofilin. This opposes nuclear import of G-actin leading to depletion of nuclear G-actin, which, in turn, activates the MRTF/SRF transcription factor to enable HGF-driven scattering and invasive behaviour of cancer cells (Fig. 8).

## Discussion

EphA2 and other ephrin receptors recruit a range of signalling adaptors to activate cascades that drive actin polymerization and other cytoskeletal rearrangements[26,38]. Direct activation of cytoskeletal rearrangements, such as increased subplasmalemmal actin dynamics[39] and the transmission and sensing of cytoskeletal tension[40], are certainly integral to the implementation of repulsive responses mediated by ephrin receptors. Nevertheless, here we provide evidence that EphA2-dependent alterations to actin dynamics influence transcriptional events in the nucleus within

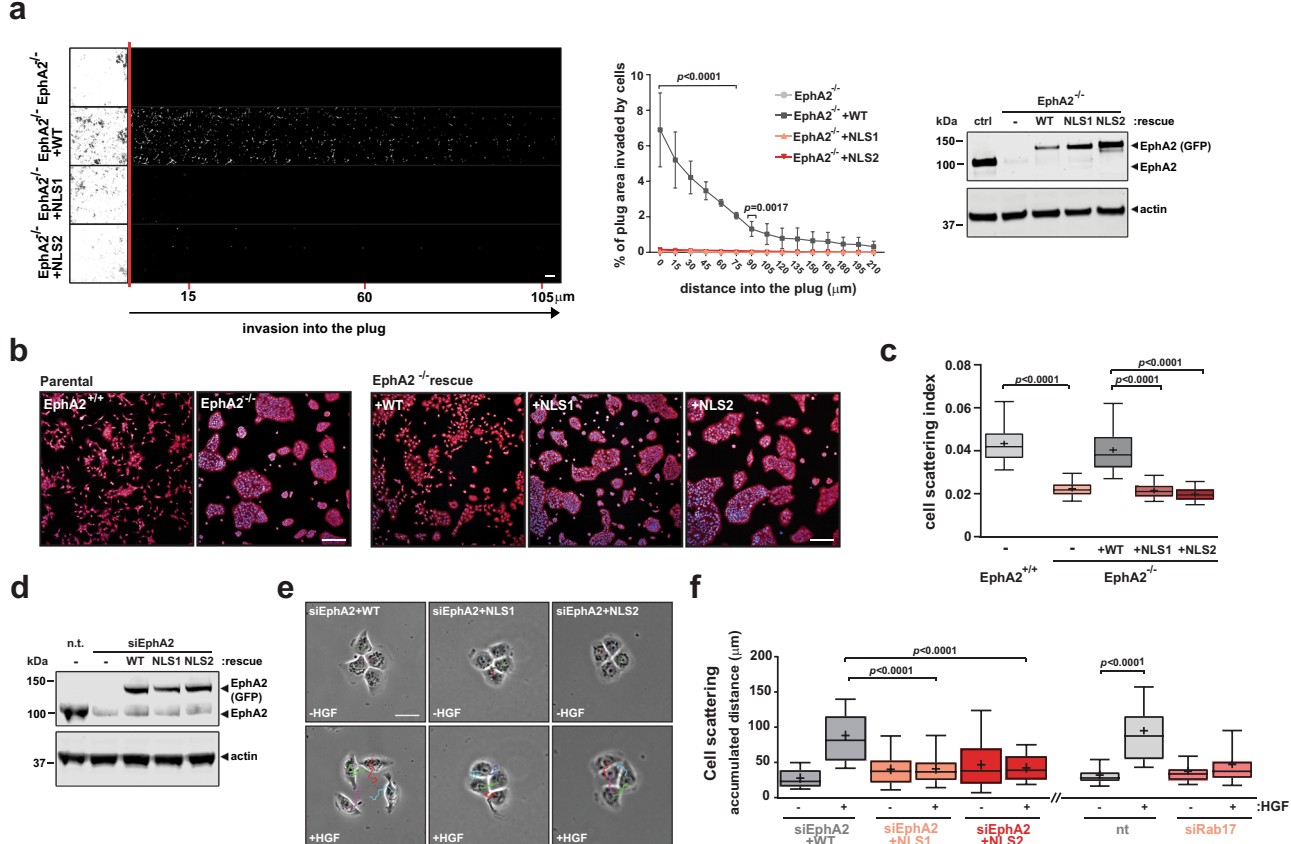

**Fig. 4 Nuclear-capture of EphA2 drives scattering and invasion in H1299 and PDAC cell lines. a–c** Primary mouse cells were derived from pancreatic adenocarcinoma (PDAC) from KPC (Pdx1-Cre, Kras$^{G12D/+}$, p53$^{R172H/+}$) mice that were either wild-type (EphA2$^{+/+}$) or knockout (EphA2$^{-/-}$) for EphA2. EphA2-knockout PDAC cells stably expressing wild-type GFP-tagged EphA2 (EphA2$^{-/-}$ + WT), or GFP-tagged EphA2s with mutated NLS sequences (EphA2$^{-/-}$ + NLS1 and EphA2$^{-/-}$ + NLS2) or empty vector control (EphA2$^{-/-}$) were generated, and expression confirmed by western blotting (**a**; right panel). Invasion of PDAC cells toward a gradient of HGF was then determined using an inverted invasion assay (**a**). Optical sections were taken every 15 μm and consecutive images are displayed as a series running from left to right (**a**; left panel). Cell invasion at the indicated distances was quantified and expressed as a % of the total quantity of fluorescent cells in the plug (**a**; centre panel) ($n = 3$ independent experiments) Values are mean ± SEM. PDAC cells expressing EphA2 and its mutants were plated onto glass surfaces and their distribution quantified using high-content imaging (**b**). Box and whiskers: 10–90 percentile whiskers, + represents mean, black line represents median. Scale bar, 300 μm. The cell scattering index (% of objects not found in clusters of ≥5 cells) is displayed in the box and whisker plot (**c**), $n = 3$ independent experiments. Statistical test is one-way ANOVA. **d–f** H1299 cells were transfected with siRNAs targeting EphA2, Rab17, or control, in combination with 'rescue vectors' containing wild-type (WT) or the indicated NLS mutants of EphA2-GFP. Knockdown and rescue of EphA2 was determined using western blotting (**d**). Cells were plated onto glass surfaces and allowed to form colonies of ~4 cells/colony and then challenged with HGF (**e**). Cell scattering was quantified using ImageJ and is expressed as the accumulated distance travelled over 8 h (**f**). Box and whiskers: 10–90 percentile whiskers, + represents mean, black line represents median, $n = 3$ independent experiments, statistical test is one-way ANOVA.

minutes of HGF addition, and that this is a pre-requisite for cell : cell repulsion to occur. Activation of transcription is known to be a key element in the early stages of certain cell : cell repulsive responses. For example, HGF has been shown to upregulate *SNAI1* and *KLF4*, which enables cell : cell repulsion and scattering by repressing expression of mRNAs for cell adhesion molecules such as E-cadherin and claudin-3[41]. In this situation, upregulation of both *SNAI1* and *KLF4* is enabled by another transcription factor, Egr-1, and this occurs within an hour of HGF addition. Egr-1 is a well-described SRF/MRTF target[42] and our data shows how its upregulation is rapidly promoted by HGF and in a way that is dependent on EphA2 expression (Fig. 5a). The acquisition of invasive cancer phenotypes cannot occur without initiation of specific transcriptional programmes. The AP-1 family of transcription factors is key to initiating invasion and it achieves this by upregulating pro-invasive genes and shutting down the expression of invasion suppressors[43]. Here we have identified two

members of the AP-1 family—*FOSB* and *JUNB*—as primary transcriptional targets of the EphA2-SRF-MRTF axis, which provides mechanistic insight into how the interplay between EphA2 and HGF may contribute to metastasis. Moreover, our observations may illuminate how alterations in the signalling coding downstream of EphA2 dictates whether cell : cell repulsion operates primarily in a tumour-suppressive or a metastasis-promoting mode. Indeed, when a cell in a normal epithelium acquires a Ras mutation, this increases EphA2 expression, which, via interaction with ephrinA ligands in the surrounding cells, activates cell contractility to promote cell : cell repulsion[44,45]. This repulsive event apically extrudes the Ras-mutated cell from the epithelium, thus enacting a tumour-suppressive role for EphA2. It will be interesting to explore how the signalling downstream of EphA2 differs between this tumour-suppressive process and the pro-invasive nuclear-capture-dependent pathway that we describe in this this study. In particular, it will be interesting to

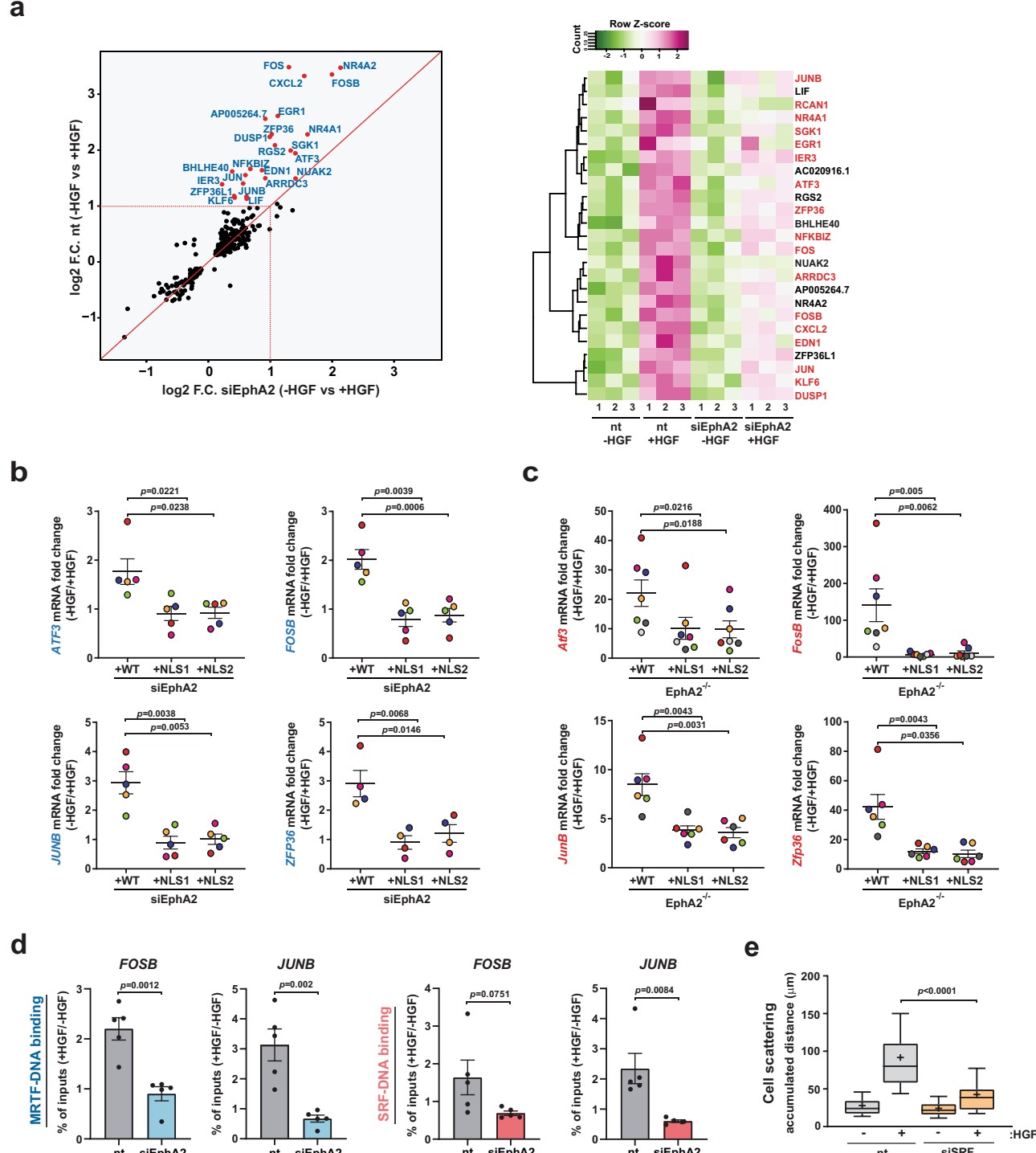

determine whether activation of MRTF/SRF-dependent transcription imparts pro-invasive/metastatic characteristics to EphA2-driven cell:cell repulsive events and whether that signalling is dependent on the balance between the levels of EphA2 expressed in the cancer cells and the abundance of ephrin ligands present in the tumour microenvironment.

The mechanisms controlling MRTF/SRF activation are likely to have evolved to coordinate gene expression with cytoskeletal reorganization and cell migration. An established route to MRTF/SRF activation is driven by increased cytosolic actin polymerization, which occurs following activation of cell migration[29,46,47]. Cell migration-associated actin polymerization reduces G-actin levels in the cytosol, which reveals MRTF's nuclear localization sequences. This then allows MRTF to enter the nucleus. Once in the nucleus, G-actin levels must be kept low within this compartment for MRTF's RPEL domains to mediate binding to its target promoter regions and enable transcription. Interestingly, selective activation of actin polymerization in the nucleus—presumably favouring reduced nuclear G-actin levels—is sufficient to trigger MRTF transcriptional activity, suggesting that it can be the nuclear (and not cytosolic) G-actin levels that are the key trigger to MRTF activation[30]. Consistent with this, we

**Fig. 5 Nuclear-capture of EphA2 drives SRF/MRTF-dependent gene expression in H1299 and PDAC cell lines. a** H1299 cells were transfected with siRNAs targeting EphA2 or control. Cells were challenged with HGF for 20 min and gene expression analysed by RNAseq. Each spot represents mRNAs that are significantly regulated by HGF. mRNAs whose HGF-dependent regulation is opposed by EphA2 knockdown are annotated. The heatmap displays the HGF-dependent regulation of the EphA2-dependent mRNAs (SRF target genes in red). $n = 3$ independent experiments. **b, c** EphA2 knockdown H1299 (siEphA2; **b**) or mouse primary PDAC cells from EphA2-knockout mice (EphA2$^{-/-}$; **c**) were transfected with wild-type EphA2 (WT) or NLS mutants of EphA2 (NLS1 and NLS2). Cells were then incubated in the presence or absence of HGF for 20 min and levels of the indicated mRNAs (FOSB; JUNB; ATF3; ZFP36) determined using qPCR. HGF-driven ($-$HGF/$+$HGF) mRNA expression fold change is plotted on the y-axis. Bars are mean ± SEM, statistical test is one-way ANOVA. In **b**, $n = 5$ for (FOSB; JUNB; ATF3) and $n = 4$ (ZFP36) independent experiments. In **c**, $n = 7$ for (Atf3; FosB) and $n = 6$ (JunB; Zfp36) independent experiments. Paired data are denoted by dots of the same colour. **d** H1299 cells were transfected with an siRNA targeting EphA2 (siEphA2) or control (nt) and challenged with HGF. Promoter-bound MRTF (left panels) or SRF (right panels) were immunoprecipitated (ChIP) and quantified by qPCR. HGF-induced fold changes in immunoprecipitated genes are plotted on the y-axis. Values are mean ± SEM, $n = 5$ independent experiments, statistical test is two-sided t-test. **e** H1299 cells were transfected with siRNAs targeting SRF (siSRF) or control (nt). Cell scattering in the presence and absence of HGF was determined as for Fig. 4f. Box and whiskers: 10–90 percentile whiskers, $+$ represents mean, black line represents median, $n = 3$ independent experiments, statistical test is one-way ANOVA.

find that depletion of nuclear (and not cytosolic) G-actin is absolutely required for HGF to trigger transcription of MRTF/SRF targets, whereas nuclear import of MRTF is triggered by HGF in a way that is independent from either nuclear or cytosolic G-actin levels. Thus, a characteristic of HGF-driven, EphA2-dependent cell:cell repulsion and invasiveness is that it operates via a mechanism in which MRTF/SRF is activated by selectively reducing nuclear (but not cytosolic) G-actin (see Figs. 6b and 7c). Our approach, involving a combination of cell biology and computational modelling approaches, has elucidated a pathway through which cells can efficiently and rapidly reduce nuclear G-actin, while maintaining (or even increasing) cytosolic G-actin, and this involves a mechanism in which endosomes are captured at the nuclear surface to drive cofilin phosphorylation at this locale. Certainly, our evidence suggests that the nuclear membrane can act as a scaffolding platform upon which EphA2-positive endosomes and the machinery downstream responsible for cofilin phosphorylation can assemble. Indeed, as EphA2s with mutated NLSs do not support HGF-driven cofilin phosphorylation, it appears that the signalling ensemble linking EphA2 to cofilin phosphorylation seems to be only fully active when EphA2 is captured at the nuclear surface. Importins are known to multitask[48]—likely contributing to processes as diverse as mitotic spindle formation[49], proteasome[50], and stress granule[51] function and microtubular transport[52]. Thus, although we have not yet demonstrated a direct association between EphA2's cytotail and an importin, we submit that NLS-mediated recruitment of EphA2-positive endosomes to the nuclear membrane, to allow signalosome assembly should be added to the list of tasks performed by elements of the nuclear import machinery. However, why does the signalosome assembled around EphA2 need to be positioned so close to the nucleus and what would be the consequences for actin dynamics if EphA2 were to activate cofilin phosphorylation further away from the nucleus? To address this, we extended our computational model to determine whether the positioning of a signalling ensemble driving cofilin phosphorylation influences nucleocytoplasmic actin dynamics. This indicated that if the region representing the congregation of nuclear-captured EphA2-positive endosomes is moved away from the nuclear surface, thus mimicking cofilin phosphorylation near to the plasma membrane, this reduces the amount of G-actin mobilized from the nucleus to the cytoplasm (Supplementary Fig. 5d). This suggests that one of the main reasons that EphA2-positive endosomes (and their associated signalling ensembles) are captured at the nuclear surface, as opposed to being assembled on endosomes further away from the nucleus, is to ensure that the HGF-driven reduction of nuclear G-actin is as efficient as possible. We propose that this would maximize activation of

MRTF$-$/SRF in the nucleus, while maintaining cytosolic G-actin levels to support the actin polymerization needed in this compartment to fully enable cell:cell repulsion and invasive cell migration to proceed.

To conclude, this study describes a new paradigm for transmission of signals between the plasma membrane and the nucleus in which one RTK (MET) promotes endocytosis of another (EphA2)—whose function is to interact with the nuclear import machinery to capture endosomes at the nuclear surface. Importantly, we have developed mathematical models, which have guided our experimental strategy, and the combination of these approaches have allowed us to conclude that EphA2-mediated nuclear-capture functions to oppose (primarily via increased cofilin phosphorylation) nuclear import of G-actin, thus driving down nuclear (but not cytoplasmic) G-actin levels to trigger SRF/MRTF-dependent gene expression. This means of signal transduction provides opportunities for coordination and integration of inputs from other pathways and we, therefore, anticipate that nuclear-captured endocytic compartments will function as a key signalling nexus for a range of cellular processes in a variety of cell types. Indeed, it has not escaped our notice that several other receptors (within the RTK and other receptor families) contain sequences within their cytodomains, which may constitute functional NLSs. Thus, it will be interesting to determine how widespread a phenomenon is nuclear-capture and the extent to which it may influence the integration of signalling downstream of other receptors controlling processes, such as T-cell development, which are known to rely on activation of MRTF/SRF-dependent transcription[53].

## Methods

**Cell culture, transfection, constructs, and siRNA**. H1299 cells were obtained from American Type Culture Collection. The genetic identity of all these cell lines was confirmed at the CRUK Beatson Institute (Promega GenePrint 10 Kit). Cell lines were cultured at 37 °C and 10% $CO_2$ in a humidified incubator. H1299 cells and in-house mouse PDAC cells were cultured in Dulbecco's modified Eagle medium. All media were supplemented with 10% fetal calf serum (FCS), 2 mM L-glutamine, 100 IU/ml penicillin, 100 μg/ml streptomycin. For expression vectors, cells were transfected using Lipofectamine 2000 (Thermo Fisher) and for siRNAs transfection was performed using the Nucleofector system (kit V; Lonza). All the cell lines used were tested for the presence of mycoplasma contamination.

PDAC cells were obtained from pancreatic adenocarcinoma from EphA2$^{+/+}$ or EphA2$^{-/-}$ KPC mice as previously described[14]. The KPC genotype is Pdx1-Cre, Kras$^{G12D/+}$, p53$^{R172H/+}$. EphA2$^{-/-}$ mice were obtained from Jackson Laboratories.

EphA2-GFP cDNA was subcloned into a pCDNA3.1-Zeo empty vector. Mutagenesis to generate the EphA2 NLS1 and NLS2 mutants was performed using the Quikchange Multi-lightning site-directed mutagenesis kit (Agilent). Importin-α5 (KPNA1) and importin-β1 (KPNB1) cDNAs were obtained from MRC PPU reagents (MRC, Dundee) and subcloned into an mCherry-N1 empty vector. mCherry-Lifeact was a gift from Dr. Laura Machesky (CRUK Beatson Institute). RhoG-GFP construct was a gift from Dr. Anne Ridley (University of Bristol). EGFP-C1-Rab17 was a gift from Jeremy Simpson (University College, Ireland). The R62D and NLS-R62D actin constructs were gifts from Dr. Robert Grosse

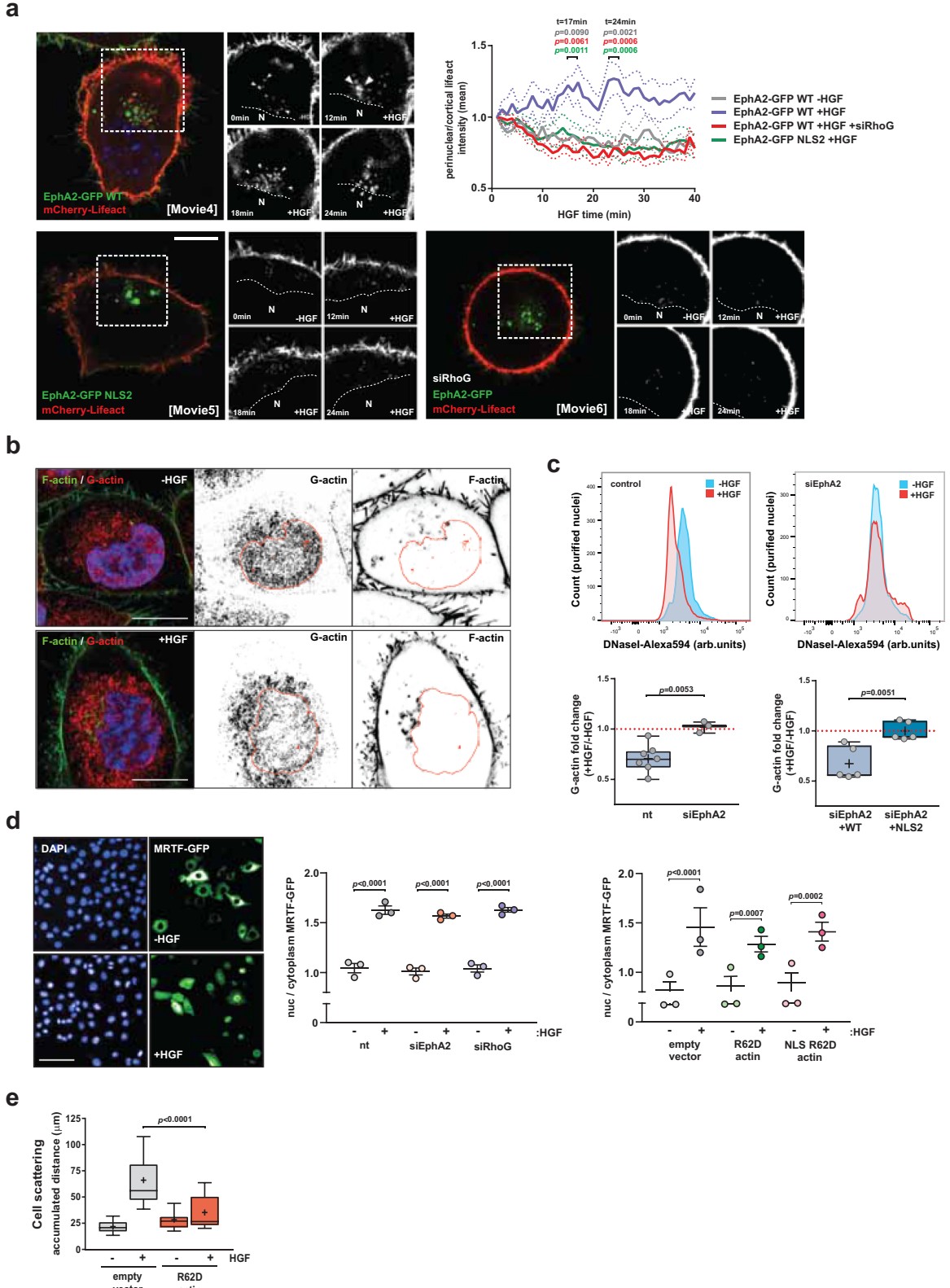

(University of Freiburg). siRNA oligos for EphA2, Rab17, SRF, RhoG and XPO6 were purchased as ON-TARGETplus siRNA SMARTpools (Dharmacon). siRNA for clathrin heavy chain is as described in ref. [54]. LIMK inhibitor BMS-5/LIMKi 3 (Tocris) was used at a 10 μM final concentration. Arp2/3 inhibitor CK-666 (Tocris) was used at a final concentration of 100 μM.

**Antibodies and immunoprecipitation.** For WB and immunofluorescence (IF), antibodies were from the following sources: goat anti-LaminA/C (Santa Cruz, sc-

6215, dilutions: WB 1 : 1000; IF 1 : 100), mouse anti-EphA2 (Millipore, 05-480, dilution: WB 1 : 1000), mouse anti-EphA2 (Santa Cruz, sc-398832, dilution: WB 1 : 500), rabbit anti-EphA2 phospho-Ser897 (Cell Signaling, #6347, dilution: WB 1 : 1000; IF 1 : 200), rabbit anti-EphA2 phospho-Tyr588 (Cell Signaling, #12677, dilution: WB 1 : 1000), mouse anti-GFP (Immunoprecipitations, Abcam, ab1218, 6.66 μg/ml), mouse anti-GFP (Santa Cruz, sc-9996, dilution: WB 1 : 500), rabbit anti-GFP (Abcam, ab6556, dilution; WB 1 : 1000), rabbit anti-importin-α5 (KPNA1, Proteintech, 18137-1-AP, dilution: WB 1 : 1000), mouse anti-importin-β1

**Fig. 6 Nuclear-capture is required for perinuclear actin polymerization and nuclear G-actin depletion. a** H1299 cells were transfected with either EphA2-GFP$^{WT}$ (WT) or EphA2-GFP$^{NLS2}$ (NLS2) (green) in combination with mCherry-Lifeact (red), with or without an siRNA targeting RhoG (siRhoG). Cells were challenged with HGF and monitored using time-lapse imaging (Supplementary Movies 4–6). Scale bar, 10 μm. Stills are presented in the inserts. Lifeact fluorescence intensity in the perinuclear region is expressed as a ratio between the perinuclear and the cortical regions. Solid and dotted lines are mean and SEM, respectively, $n > 7$ independent experiments, statistical test is two-way ANOVA and *p*-values are comparisons (at the indicated time points) between the EphA2-GFP WT + HGF values and other conditions as denoted by the colours. **b** Cells were incubated in the presence or absence of HGF for 20 min, fixed and stained with fluorescent DNaseI to visualize G-actin (G-actin; red) and counterstained with phalloidin (green) and DAPI (blue). The nucleus is denoted in red. Scale bar, 10 μm. **c** Cells were transfected with siRNAs targeting EphA2 (siEphA2) or control (nt), or siEphA2 in combination with wild-type EphA2-GFP (WT) or EphA2 NLS2-GFP (NLS2). Cells were challenged with (+HGF) or without HGF (−HGF) for 20 min. Nuclear G-actin content was quantified by flow cytometry. Box and whisker plots (min. to max. whiskers, + represents mean) are ratio of the geometrical mean value of HGF-treated vs. non-treated cells (+HGF/−HGF), $n = 7$ (control, left panel), $n = 3$ (siEphA2, left panel), and $n = 5$ (right panel) independent experiments, statistical test is unpaired *t*-test two-sided analysis. **d** Cells expressing MRTF-GFP were transfected with siRNAs targeting EphA2, RhoG or R62D, or nuclear-targeted R62D actin mutants or vector control, and challenged with HGF as indicated. Distribution of MRTF-GFP was expressed as the ratio of MRTF-GFP in the nucleus vs. the cytoplasm. Values are expressed as mean ± SEM, $n = 3$ independent experiments, statistical test is two-way ANOVA. **e** Cells were transfected with R62D actin mutant or vector control. Scattering was determined as for Fig. 2c. Box and whiskers: 10–90 percentile whiskers, + represents mean, black line represents median, $n = 3$ independent experiments, statistical test is one-way analysis ANOVA.

(KPNB1, Cell Signaling, #60769, dilution: WB 1 : 1000), mouse anti-actin (Sigma, A1978, dilution: WB 1 : 5000), mouse anti-RhoG (Millipore, 04-486, dilution: WB 1 : 1000), mouse anti-Rab17 (Abnova, H00064284-MO1, dilution: WB 1 : 1000), mouse anti-importin-α7 (Proteintech, 12366-2-AP, dilution: WB 1 : 1000), rabbit anti-TNPO3 (Abcam, ab109386, dilution: WB 1 : 1000), mouse anti-CHC (TD1, F. Brodsky, dilution: WB 1 : 1000), rabbit anti-Histone H2a (Abcam, ab16563, dilution: WB 1 : 1000) and rabbit anti-XPO6 (Proteintech, 11408-1-AP, dilution: WB 1 : 1000), mouse anti-cofilin (Proteintech, 66057-1-Ig, dilution: WB 1 : 5000) and rabbit anti-cofilin phospho-Ser3 (Santa Cruz, sc-12912-R, dilution: WB 1 : 1000). For immunoprecipitation, mouse antibodies were coupled to magnetic beads conjugated to anti-mouse IgG (Invitrogen; Dynabeads Sheep anti-mouse IgG; catalogue number 11031). For chromatin immunoprecipitation, antibodies were rabbit anti-SRF (Cell Signaling Technologies, #5147, 5 μg/immunoprecipitation), rabbit anti-MRTF-A (Cell Signaling Technologies, #14760, 5 μg/immunoprecipitation) and mouse anti-ELK-1 (Santa Cruz, sc-365876, 5 μg/immunoprecipitation). For the TurboID experiments, we used mouse anti-RanBP2 (Santa Cruz, sc-74518, dilution: WB 1 : 500), rabbit anti-Nup214 (Bethyl Laboratories, A300-716A-M, dilution: WB 1 : 300), rabbit anti-POM121 (Genetex, GTX102128, dilution: WB 1 : 1000), mouse anti-Nup133 (Santa Cruz, sc-376763, dilution: WB 1 : 500), mouse anti-Nup88 (BD Biosciences, 611896, dilution: WB 1 : 1000), mouse anti-Nup50 (Santa Cruz, sc-398993, dilution: WB 1 : 250), goat anti-laminA/C (Santa Cruz, sc-6215, dilution: WB 1 : 1000) and mouse anti-V5-Tag (Thermo, MA5-15253, dilution: WB 1 : 1000; IF 1 : 100). Cell lysates were prepared in a buffer containing 200 mM NaCl, 75 mM Tris-HCl pH 7, 15 mM NaF, 1.5 mM Na$_3$VO$_4$, 7.5 mM EDTA, 7.5 mM EGTA, 0.15% (v/v) Tween-20, and protein inhibitors (Thermo Fisher). Lysates were passed three times through a 26-gauge needle and clarified by centrifugation at 10,000 × *g* for 5 min at 4 °C. Lysates were added to the beads and rotated for 2 h at 4 °C. Beads were washed three times in Tween-20-containing buffer and then analysed by WB.

**Quantitative reverse-transcription PCR.** Trizol (Ambion) was used to isolate total RNA from the relevant cell lines following the manufacturer's protocol. The cDNA was obtained by using the Quantitect reverse-transcription kit (Qiagen). Quantitative reverse-transcription PCR (qRT-PCR) reactions were prepared using the SYBR Green kit (QuantaBio). The amplified products were obtained and analysed by a CFX96 qPCR System (BioRad) and BioRad CFX Manager 3.1 software. ΔΔC(*t*) was determined using ribosomal *18S* as a reference. Control transfected transcript levels were assigned the arbitrary value of 1. *FOSB*, *JUNB*, *ATF*3, *ARHGEF16* and *RAB17* human Quantitect primers were purchased from Qiagen and the rest of the primers were manufactured by Invitrogen (Thermo Fisher) (see Supplementary table 1).

**Quantification of EphA2 internalization and nuclear-capture**
*Internalization.* Cell surface proteins were labelled with membrane impermeant sulfo-NHS-SS-Biotin (0.13 mg/ml) in phosphate-buffered saline (PBS) for 30 min at 4 °C. To allow internalization, cells were incubated at 37 °C for the appropriate times in the presence and absence of HGF (10 ng/ml, Preprotech). To remove biotin from proteins remaining at the cell surface, cells were incubated with sodium mercaptoethanesulphonate (MesNa; 20 mM) for 60 min at 4 °C, and the cell lysed in a buffer containing 1.5% Triton X-100 and 0.75% NP-40 (pH 7.0). Biotinylated (internalized) EphA2 was then determined by capture-ELISA. Maxisorb 96-well plates (Life Technologies) were coated overnight with 5 μg/ml anti-EphA2 antibodies (Millipore) in 0.05 M Na$_2$CO$_3$ pH 9.6 at 4 °C and blocked in PBS containing 0.05 % Tween-20 (PBS-T) with 5% bovine serum albumin (BSA) for 1 h at room temperature. EphA2 was captured by overnight incubation of 50 μl of cell lysate at

4 °C. Unbound material was removed by extensive washing with PBS-T and wells were incubated with streptavidin-conjugated horseradish peroxidase (Vector Laboratories) in PBS-T containing 1% BSA for 1 h at 4 °C. Following further washing, biotinylated integrins were detected by chromogenic reaction with ortho-phenylenediamine followed by measurement of absorbance at 490 nm in a multi-well spectrophotometer (Magellan software 7.2 – Tecan)[55].

*Nuclear-capture.* Cells were collected in PBS, centrifuged for 10 s in a tabletop centrifuge and the supernatant discarded. Cells were resuspended in PBS containing 0.1% NP-40 (PBS-N) and passed through a 26-gauge needle 5 times, centrifuged for 10 s and resuspended in PBS-N. This procedure was then repeated. The pellet was finally resuspended in lysis buffer containing 1% Triton X-100 and 1% NP-40, sonicated for three rounds of 20 s/round. Finally, samples were centrifuged at 6800 × *g* in a tabletop centrifuge for 30 s and protein content was measured by using Optiblot (Thermo Fisher) following the manufacturer's protocol. The levels of nuclear-captured biotinylated-EphA2 were determined by capture-ELISA as described above using ELISA plates coated with either mouse anti-EphA2 or mouse anti-GFP (Abcam) antibodies. To image purified nuclei by IF, nuclei were seeded onto glass-bottom dishes previously coated with poly-D-lysine, fixed with 4% paraformaldehyde and stained with various antibodies.

To image internalization of biotinylated proteins, the same cell surface biotinylation and MesNa reduction procedure as described above was used followed by fixation in 4% paraformaldehyde. Biotinylated proteins were visualized with Alexa488-conjugated streptavidin (Vector). Images were acquired using a Zeiss LSM 880 Airyscan confocal microscope and processed using Zen Black Zeiss software (version 2.3.sp1). To quantify distances between vesicles and the nucleus in ImageJ software (version 1.53), a mask for the 4′,6-diamidino-2-phenylindole (DAPI) staining was created and a distance map obtained from the boundaries of the nucleus. Internalized particles were selected and the distance to the nucleus was analysed by overlaying them with the distance map. The analysis was performed by obtaining ten fields of view (FoV) for each condition in each independent experiment, in which we measured an average of three cells/FoV and analysed an average of 32 particles per cell.

**Cell scattering and invasion**
*Cell scattering.* H1299 cells were seeded onto 6-well plates for 48 h (65,000 cells per well), during which time the cells formed small colonies. Cells were then visualized using a Nikon time-lapse microscope in the presence and absence of HGF (10 ng/ml), with or without CK-666 (100 μM) or BMS-5 (10 μM). Images were collected every 5 min from six different regions per well (Metamorph software— version 7.8.13.0). To track scattering, ImageJ manual tracking and chemotaxis plugins were used.

*Invasion.* Matrigel was allowed to polymerize in Transwell inserts (Corning) for 1 h at 37 °C. Inserts were then inverted and cells seeded directly onto the upper face of the filter. Cells were placed in a chemotactic gradient of medium supplemented with 10% FCS and 10 ng/ml HGF, and, 48 h after seeding, migrating cells were stained with Calcein-AM and visualized by confocal microscopy with serial optical sections being captured at 15 μm intervals[56].

**High-content image analysis**
*Cell scattering.* PDAC cells were seeded onto glass six-well plates. After 24 h, cells were fixed and stained with DAPI (Sigma) and Cell Mask (1 : 10,000, Invitrogen). Cells were imaged at ×10 magnification using the Opera Phenix High-Content

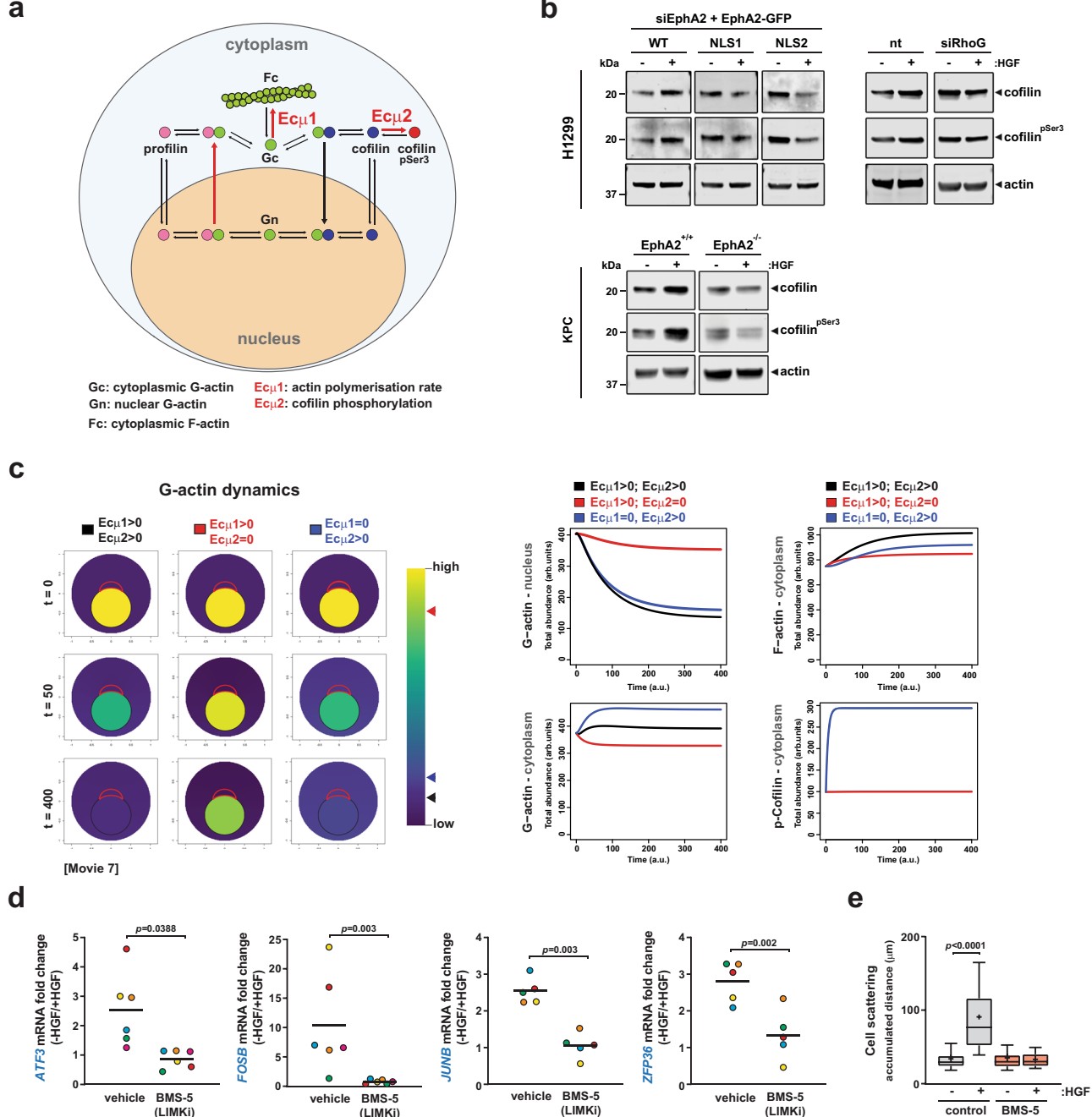

**Fig. 7 Cofilin phosphorylation controls nucleocytoplasmic actin dynamics and gene expression. a** Schematic of pathways downstream of EphA2 influencing nucleocytoplasmic actin dynamics. Rates of cytoplasmic actin polymerization and cofilin phosphorylation are denoted by $E_c\mu1$ and $E_c\mu2$, respectively. **b** H1299 cells were transfected with siRNAs targeting RhoG (siRhoG) or control (nt), and EphA2 knockdown H1299 cells were transfected with rescue vectors for wild-type EphA2 (WT), EphA2 NLS1 (NLS1) or EphA2 NLS2 (NLS2). PDAC-derived cell lines were from KPC mice (KPC EphA2[+/+]) or EphA2-knockout KPC mice (KPC EphA2[−/−]). Levels of cofilin and phospho-cofilin (cofilin pSer[3]) following HGF treatment were determined by western blotting. **c** Nucleocytoplasmic actin dynamics were modelled according to the schematic in **a**. The cell is modelled as two static circular compartments representing the cytoplasm and nucleus (see Supplementary Fig. 5a). The area of influence of nuclear-captured EphA2 is delimited with a red line and G-actin levels are represented by the Viridis colour palette. EphA2-dependent actin polymerization ($E_c\mu1$) and cofilin phosphorylation ($E_c\mu2$) were 'initiated' at $t = 0$ and movies made from this (Supplementary Movie 7). Stills (at $t = 0$, 50 and 400) from these movies and graphs denoting the changes in G-actin (nuclear and cytoplasmic), F-actin and phospho-cofilin levels are displayed. This dynamic model indicates that shutdown of cofilin phosphorylation ($E_c\mu1 > 0$; $E_c\mu2 = 0$; red), but not actin polymerization ($E_c\mu1 = 0$; $E_c\mu2 > 0$; blue), opposes depletion of nuclear G-actin. **d** Cells were treated with or without HGF for 20 min in the absence or presence of LIMK inhibitor (BMS-5) and levels of the indicated SRF target determined as for Fig. 5b. Bar is mean of $n = 6$ for (*ATF3*; *FOSB*) and $n = 5$ (*JUNB*, *ZFP36*) independent experiments. Paired data are denoted by dots of the same colour. **e** Cell scattering was determined as for Fig. 4f, in the presence and absence of LIMK inhibitor (BMS-5). Box and whiskers: 10–90 percentile whiskers, + represents mean, black line represents median, $n = 3$ independent experiments, statistical test is one-way ANOVA.

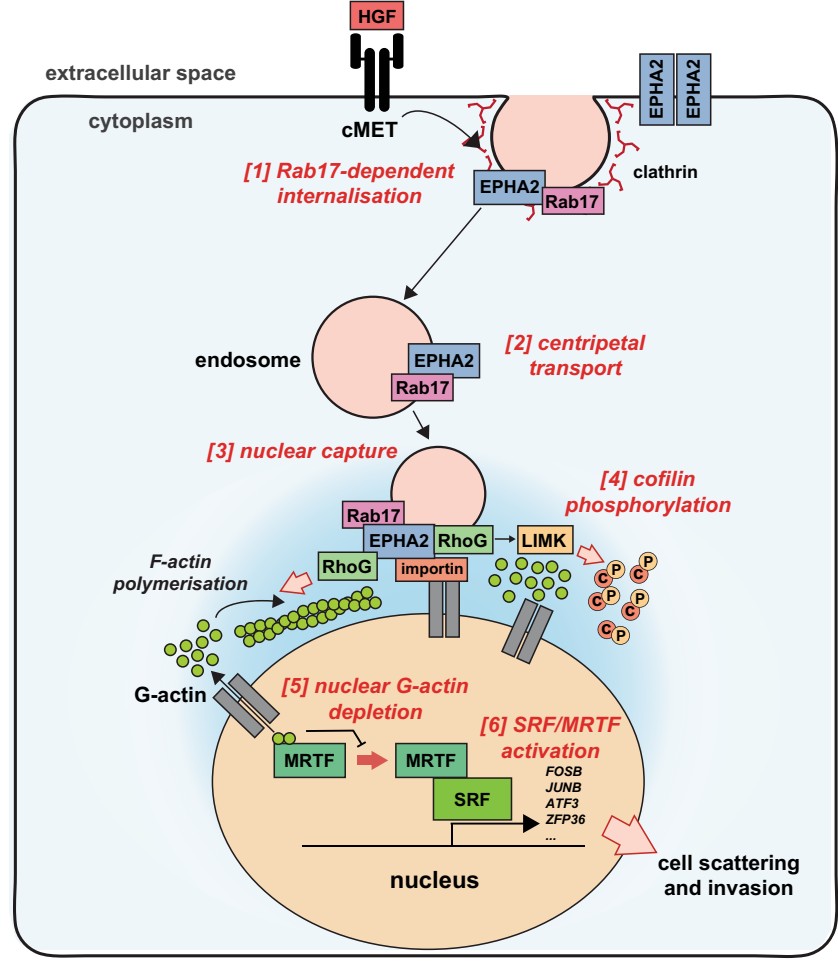

**Fig. 8 Schematic summary of the role played by nuclear-capture of EphA2 endosomes in controlling nuclear G-actin dynamics.** MET drives clathrin and Rab17-dependent endocytosis of EphA2 [1]. EphA2/Rab17-positive endosomes are then transported centripetally [2] and become physically attached to, or 'captured' by, the nucleus by an interaction formed between the nuclear import machinery and a nuclear localization sequence located in EphA2's cytodomain [3]. This nuclear-capture event, in turn, drives actin polymerization, which is restricted to the juxta-nuclear region [4] and LIMK-driven phosphorylation of cofilin [5], and both events are dependent on the RhoG GTPase. Phosphorylation of cofilin opposes nuclear import of cofilin–actin, leading to depletion of G-actin from the nucleus [6], which, in turn, activates transcription of MRTF/SRF target genes [7] to implement cell scattering and invasion.

Screening System (Harmony High-content Imaging and analysis software version 4.9, PerkinElmer) and cell distribution was quantified using Columbus Image Data Storage and Analysis System (PerkinElmer version 2.8.0). A cell cluster was defined as five or more nuclei <1 μm from each other with an area >600 μm$^2$.

*Nucleocytoplasmic distribution of MRTF-GFP.* H1299 cells stably expressing an MRTF-GFP construct (gift from Dr. Robert Grosse) were plated in six-well dishes and treated in the presence or absence of HGF. Cells were processed and imaged as described before. MRTF localization was quantified by using DAPI as a nuclear mask and DAPI-negative Cell Mask staining for the cytoplasmic mask.

**RNA sequencing.** RNA was extracted as described in the qRT-PCR methods. Quality of the purified RNA was tested on a 2200 Tapestation using RNA screentape (Agilent). Libraries for cluster generation and DNA sequencing were prepared using the TruSeq Stranded mRNA LT Kit (Illumina). Quality and quantity of the DNA libraries was assessed on the 2200 Tapestation (D1000 screentape) and Qubit (Thermo Fisher Scientific), respectively. The libraries were run on the Next Seq 500 (Illumina) using the High Output 75 cycles kit (2 × 36cycles, paired-end reads, single index). To analyse the RNAseq expression data, quality checks on the raw RNASeq data files were performed using fastqc version 0.11.7 and fastq screen version 0.12.0. RNAseq paired-end reads were aligned to the GRCm38 version of the human genome and annotation using HiSat2

version 2.1.0. Expression levels were determined and statistically analysed using a combination of HTSeq version 0.9.1, the R environment version 3.4.4, utilizing packages from the Bioconductor data analysis suite and differential gene expression analysis was performed using voom pipeline from the limma package in R.

**Chromatin immunoprecipitation.** H1299 cells were incubated in the presence or absence of HGF and cross-linked by adding paraformaldehyde (final concentration 1%) for 10 min at room temperature. Glycine (125 mM) was added and incubated for further 5 min. Collected cells were spun and washed twice in PBS and once in PBS/NP-40. Pellets were incubated for 30 min in ice-cold high-salt buffer and hypotonic disruption was then performed by incubating the pellets in low-salt buffer for 30 min. Samples were mechanically disrupted with a 26 G syringe, centrifuged and resuspended in low-salt buffer and 20% sarkosyl. DNA pellets were obtained by centrifuging the samples on a sucrose cushion. Sonication was performed in a 30 s-on 30 s-off manner for 30 min at 4 °C and the resulting DNA was quantified using Qubit ds HS DNA kit (Invitrogen). Ten percent of each sample was kept as inputs and 25 μg of DNA were used for immunoprecipitation using antibodies recognizing either MRTF, SRF or Elk-1 (5 μg per sample) and magnetic beads (MagnaBind Goat anti-rabbit IgG or Dynabeads Sheep anti-mouse, Thermo Scientific). Immunoprecipitation was performed overnight at 4 °C and samples were washed in radioimmunoprecipitation assay (RIPA) buffer containing LiCl buffer and then in a Tris-EDTA buffer. Eluted samples were treated with

Proteinase K and cleaned using a PCR clean up kit (Qiagen). The final eluates were quantified by qPCR.

## Proteomics for determination of the plasma membrane to nucleus traffickome

*SILAC labelling and isolation of cell surface and nuclear proteomes.* To obtain the surface proteome, H1299 cells were SILAC-labelled with heavy (Cambridge Isotope Laboratories) and light (Sigma) amino acids. SILAC-labelled cells were plated into 15 cm plastic dishes and, 48 h later, were surface-biotinylated using sulpho-NHS-SS-Biotin (0.15 mg/ml) in PBS at 4 °C for 1 h. Heavy-labelled cells were incubated with MesNa in Tris-buffered saline (pH 8.6) for 50 min at 4 °C to remove biotin from the cell surface, whereas the light-labelled cells remained unreduced. Cells were lysed and lysates from the heavy and light SILAC-labelled cells were mixed, and biotinylated proteins were captured by incubation with streptavidin–agarose beads (Upstate, Millipore) for 1.5 h at 4 °C with constant rotation. Following extensive washing, biotinylated proteins were eluted from the beads by reduction with dithiothreitol (DTT, 0.1 M) in a Tris buffer (pH 7.5). To determine the proteins that translocate from the cell surface to the nucleus, cells were labelled with light or heavy SILAC amino acids and surface biotinylated. Light-labelled cells were warmed to 37 °C for 30 min to allow internalization of labelled protein in the presence of HGF (10 ng/ml), whereas the heavy-labelled cells were warmed to 37 °C in the absence of HGF. Biotin remaining at the cell surface was removed by surface reduction as above. Cells were washed in PBS and nuclei were purified as described above. The supernatants from the heavy and light SILAC-labelled cells were then mixed and biotinylated proteins isolated using streptavidin-conjugated agarose beads as described above.

*Mass spectrometry.* Proteins were separated on 4–12% gradient NuPAGE Novex Bis-Tris gel (Life Technologies) and visualized using Instant blue (Expedeon). Each gel lane was excised into six slices, which were reduced using 10 mM dithiothreitol, alkylated with 55 mM iodoacetamide, and digested with trypsin (Trypsin gold, Promega), overnight at 35 °C. Tryptic peptides were desalted and dried in a centrifugal evaporator. Dried tryptic peptides were resuspended in and loaded with a buffer containing 2% acetonitrile and 0.1% formic acid, and separated using a 20 cm fused silica emitter (New Objective) packed in-house with reverse-phase Reprosil-Pur Basic 1.9 μm (Dr. Maisch GmbH). Tryptic peptides were analysed on an Orbitrap Q-Exactive HF mass spectrometer (Thermo Fisher Scientific) coupled online to an EASY-nLC II (Thermo Fisher Scientific). For the full scan a resolution of 60,000 at 250 Th was used. The top ten most intense ions in the full MS were isolated for fragmentation with a target of 50,000 ions at a resolution of 15,000 at 250 Th. MS data were acquired using the XCalibur software (Thermo Fisher Scientific - Version 4.2.28.12).

*Analysis of MS data.* MS files were processed with MaxQuant software[57] version 1.5.5.1 and searched with the Andromeda search engine[58] against the human UniProt database (09/07/2016; 92,939 entries). Common reverse and contaminant hits (as defined in MaxQuant output) were removed. Only protein groups identified with at least one uniquely assigned peptide were used for the quantification. For the SILAC experiments, the SILAC ratios between light-isotope and medium-isotope peptides were calculated using MaxQuant. Protein groups were considered reproducibly quantified if identified and quantified in both the surface proteome and the nuclear-capture proteome

## Proteomics for determination of the EphA2 proximity interactome

*TurboID construct.* Human EphA2 was cloned into a TurboID expression construct to generate the EphA2-V5-TurboID-T2A-blue fluorescent protein (BFP) fusion protein with either wild-type EphA2 or the NLS mutants. This construct allows specific identification using the V5 tag and contains a short flexible linker between EphA2-V5 and the TurboID region. The T2A self-cleaving peptidic sequence ensures the equimolar expression of EphA2-V5-TurboID and BFP, while allowing the release of BFP from the main construct into the cytoplasm. The EphA2-V5-TurboID-T2A-BFP construct was subsequently inserted into a lentiviral expression vector and viruses were generated as detailed before[59]. To generate stable cell lines, H1299 cells were transduced and cells were first selected based on the expression of BFP and then further selected using blasticidin (Invitrogen).

*Pull-down of biotin-labelled proteins.* Cells were incubated for 1 h with 50 μM biotin (Sigma) and 10 ng/ml of HGF (Peprotech). Next, cells were collected and lysed using a RIPA lysis buffer (containing 1% NP-40) supplemented with protease and phosphatase inhibitors (Thermo). To remove the excess of intracellular free-biotin and avoid the subsequent saturation of the streptavidin beads, lysates were filtered by centrifugation using a 3 K Amicon Ultra filter column (0.5 ml, Merck). Filtered lysates were incubated with streptavidin–agarose beads (Merck) for 2 h at 4 °C on a rotating wheel. After the incubation, the beads were washed using 2× RIPA, 1× KCl 1 M, 1× 0.1 M $Na_2CO_3$ and 1× 2 M urea (Tris-HCl pH 8.0). After the last wash, the beads were covered with a layer of storage solution (2 M urea in 100 mM $NH_4HCO_3$) and frozen until processed for MS. For WB analysis, purified biotin-labelled proteins were eluted from the beads with electrophoresis sample buffer containing DTT and separated by gel electrophoresis before membrane transfer.

*On-beads proteolytic digestion.* The purified proteins were digested on the beads[60] and desalted using StageTip[61].

*Mass spectrometry.* Desalted tryptic peptides were separated by nanoscale C18 reverse-phase liquid chromatography using an EASY-nLC II 1200 (Thermo Fisher Scientific) coupled to an Orbitrap Fusion Lumos mass spectrometer (Thermo Fisher Scientific). Samples were loaded into a 50 cm fused silica emitter (New Objective) packed in-house with ReproSil-Pur C18-AQ, 1.9 μm resin (Dr Maisch GmbH). The emitter was kept at 50 °C by means of a column oven (Sonation) integrated into the nanoelectrospray ion source (Thermo Fisher Scientific). Elution was carried out at a flow rate of 300 nl/min using a binary gradient with buffer A (2% acetonitrile) and B (80% acetonitrile), both containing 0.1% formic acid. An Active Background Ion Reduction Device (ESI Source Solutions) was used to decrease air contaminants signal level. Data were acquired using Xcalibur software (Thermo Fisher Scientific), with a mass spectrometer operating in data-dependent acquisition in positive ion mode. A full scan was acquired over mass range of 350–1400 $m/z$, with 60,000 resolution at 200 $m/z$, for a maximum injection time of 50 ms and with a target value of 5e5 ions with MonoIsotopic Precursor Selection set to "Peptide" mode. Higher energy collisional dissociation fragmentation was performed on the 15 most intense ions, for a maximum injection time of 100 ms, or a target value of 75,000 ions. Peptide fragments were analysed in the Orbitrap at 15,000 resolution.

*Proteomics data analysis.* The MaxQuant software[57] version 1.6.14.0 was used to process MS Raw files and searched with Andromeda search engine[58], querying UniProt[62] Homo sapiens database (09/07/2016; 92,939 entries). Specificity for trypsin cleavage and maximum two missed cleavages were requested for the search. Methionine oxidation, di-Gly-lysine and N-terminal acetylation were specified as variable modifications, and Cysteine carbamidomethylation as fixed modification. The peptide and protein false discovery rate (FDR) was set to 1%. MaxQuant outputs were analysed with Perseus software[63] version 1.6.0.7 and 1.6.2.3. The common reverse and contaminant hits (as defined in MaxQuant output) were removed. Only protein groups identified with at least one uniquely assigned peptide were used for the quantification. For label-free quantification (LFQ), proteins quantified in all five replicates in at least one group, were measured according to the LFQ algorithm available in MaxQuant[64]. Data presented in the volcano plots were expressed by using the 'Student's t-test difference' metric, which corresponds for a given protein, to the average of the log2-transformed LFQ intensities measured in each replicate from condition 1 (WT), subtracted from the counterpart average in condition 2 (either NLS1 or NLS2): e.g., [average log2-transformed LFQ intensity in condition A] − [average log2-transformed LFQ intensity in condition B]. All protein intensities measured in each replicate experiment were normalized against the corresponding EphA2 protein (bait) intensity. Missing values were imputed separately for each column (width 0.3, down shift 1.8) and significantly enriched proteins were selected using a t-test with a 5% FDR (permutation based).

**Live-imaging microscopy.** Cells expressing EphA2-GFPs in combination with either mCherry-Lifeact or Rab17-mCherry were cultured on glass-bottom 35 mm dishes. To obtain time-lapse movies, the cells were imaged in a Zeiss LSM 880 Airyscan confocal microscope at 37 °C with controlled humidity and $CO_2$. The first images were obtained in the absence of HGF and then cells were treated with HGF (10 ng/ml) and imaged at time intervals of 1 min for 40 min. Raw images were processed and subjected to deconvolution using Zen Black Zeiss software.

For quantification of actin polymerization, deconvolved images were analysed in ImageJ software by measuring theintensity value for the mCherry-Lifeact signal in each time point in a region of interest that comprised the perinuclear internalised EphA2-GFP-positive vesicles. F-actin in the perinuclear region was then expressed as a ratio of the mean intensity value of the perinuclear region versus the mean intensity value in the cortical actin region.

**GSEA enrichment analysis.** Enrichment plots were generated using GSEA 4.1.0 software (Broad Institute). Data files were generated with the data obtained in the proteomics analysis for the EphA2-TurboID interactome. The datasets were processed using GSEA software by running a pre-ranked analysis against the GO:term 0005643, corresponding to the cellular component 'nuclear pore component'.

**Quantification of nuclear G-actin.** Purified nuclei were resuspended in PBS, fixed in 4% paraformaldehyde for 20 min and permeabilized with 0.2% Triton X-100 in PBS for 10 min at 37 °C. Nuclei were stained with Alexa594-conjugated DNAseI (0.3 μM) in combination with DAPI for 1 h 30 min at 37 °C under continuous shaking, followed by resuspension in PBS containing 1 mM EDTA and 1 mM $MgCl_2$. Data were acquired in LSRFortessa flow cytometer (BD FACSDIVA software, version 8.0.1) and analysed with the FlowJo software (version 10.1r5). Geometrical mean value corresponding to the DAPI positive population was calculated, and data were expressed as the ratio of the geometrical mean value of HGF-treated cells vs. vehicle-treated cells.

**Computational model to predict nuclear G-actin dynamics**. We model the cell as two static circular compartments, $\Omega_c$ and $\Omega_n$, which represent the cytoplasm and the nucleus, respectively,

$$\Omega_C = \{ \boldsymbol{x} \in \mathbb{R}^2 : \| \boldsymbol{x} - \boldsymbol{p}_1 \| > r_1 \cap \| \boldsymbol{x} - \boldsymbol{p}_2 \| > r_2 \}, \tag{1}$$

$$\Omega_n = \{ \boldsymbol{x} \in \mathbb{R}^2 : \| \boldsymbol{x} - \boldsymbol{p}_1 \| < r_1 \}, \tag{2}$$

for some points $\boldsymbol{p}_1$ and $\boldsymbol{p}_2$, and some radii $r_1$ and $r_2$. We also let $\Gamma_c$ and $\Gamma_n$ denote the external boundary of the cytoplasm and the nucleus : cytoplasm boundary, respectively,

$$\Gamma_c = \{ \boldsymbol{x} \in \mathbb{R}^2 : \| \boldsymbol{x} - \boldsymbol{p}_2 \| = r_2 \}, \tag{3}$$

$$\Gamma_n = \{ \boldsymbol{x} \in \mathbb{R}^2 : \| \boldsymbol{x} - \boldsymbol{p}_1 \| = r_1 \}. \tag{4}$$

A graphical illustration of the simulation domain is provided in Supplementary Fig. 5a.

It is assumed that the cytoplasm contains a population of G-action, $G_c(\boldsymbol{x},t)$, which diffuses and polymerizes to form F-action, $F_c(\boldsymbol{x},t)$, at a rate of $\gamma G_c$. In the presence of a catalytic growth factor, $E_c(\boldsymbol{x},t)$, $G_c(\boldsymbol{x},t)$ polymerizes to form $F_c(\boldsymbol{x},t)$ at a rate of $\gamma_{G_c}(1 + \mu_1 E_c)$. $F_c(\boldsymbol{x},t)$ depolymerizes at a rate of $\gamma_{F_c}$ to form $G_c(\boldsymbol{x},t)$. We also assume that the cytoplasm contains a population of cofilin, $C_c(\boldsymbol{x},t)$, which associates with $G_c(\boldsymbol{x},t)$ at a rate of $\alpha_{\Xi_c}$ to form cofilin–actin, $\Xi_c(\boldsymbol{x},t)$, and dissociates from $\Xi_c(\boldsymbol{x},t)$ at a rate of $\gamma_{\Xi_c}$. In addition, $C_c(\boldsymbol{x},t)$ is assumed to phosphorylate at a rate of $\alpha_{\Theta_c}(1 + \mu_2 E_c)$, giving phospho-cofilin, $\Theta_c(\boldsymbol{x},t)$, which dephosphorylates at a rate of $\gamma_{\Theta_c}$. Finally, we assume that the cytoplasm contains a population of profilin, $P_c(\boldsymbol{x},t)$, which associates with $G_c(\boldsymbol{x},t)$ at a rate of $\alpha_{\Upsilon_c}$ to form profilin–actin, $\Upsilon_c(\boldsymbol{x},t)$, and dissociates from $\Upsilon_c(\boldsymbol{x},t)$ at a rate of $\gamma_{\Upsilon_c}$. The governing equations in the cytoplasm are therefore

$$\frac{\partial G_C}{\partial t} = D_{G_c} \nabla^2 G_c + \gamma_{F_c} F_c - \gamma_{G_c}(1 + \mu_1 E_c)G_c + \gamma_{\Xi_c}\Xi_c - \alpha_{\Xi_c}G_cC_c + \gamma_{\Upsilon_c}\Upsilon_c - \alpha_{\Upsilon_c}G_cP_c, \tag{5}$$

$$\frac{\partial F_c}{\partial t} = D_{F_c} \nabla^2 F_c - \gamma_{F_c}F_c + \gamma_{G_c}(1 + \mu_1 E_c)G_c, \tag{6}$$

$$\frac{\partial C_c}{\partial t} = D_{C_c} \nabla^2 C_c - \alpha_{\Xi_c}G_cC_c + \gamma_{\Xi_c}\Xi_c - \alpha_{\Theta_c}(1 + \mu_2 E_c)C_c + \gamma_{\Theta_c}\Theta_c, \tag{7}$$

$$\frac{\partial \Xi_c}{\partial t} = D_{\Xi_c} \nabla^2 \Xi_c - \gamma_{\Xi_c}\Xi_c + \alpha_{\Xi_c}G_cC_c, \tag{8}$$

$$\frac{\partial P_c}{\partial t} = D_{P_c} \nabla^2 P_c - \alpha_{\Upsilon_c}G_cP_c + \gamma_{\Upsilon_c}\Upsilon_c, \tag{9}$$

$$\frac{\partial \Upsilon_c}{\partial t} = D_{\Upsilon_c} \nabla^2 \Upsilon_c - \gamma_{\Upsilon_c}\Upsilon_c + \alpha_{\Upsilon_c}G_cP_c, \tag{10}$$

$$\frac{\partial \Theta_c}{\partial t} = D_{\Theta_c} \nabla^2 \Theta_c - \gamma_{\Theta_c}\Theta_c + \alpha_{\Theta_c}(1 + \mu_2 E_c)C_c, \tag{11}$$

$$\frac{\partial E_c}{\partial t} = D_{E_c} \nabla^2 E_c - \gamma_{E_c}, \tag{12}$$

for $\boldsymbol{x} \in \Omega_c$, where $D_{[\cdot]}$ denotes the diffusion coefficient of species $[\cdot]$.

In the nucleus, we assume that there exists a population of nuclear G-actin, $G_n(\boldsymbol{x},t)$, and a population of cofilin, $C_n(\boldsymbol{x},t)$, which associate with each other at a rate of $\alpha_{\Xi_n}$ to form nuclear cofilin–actin, $\Xi_n(\boldsymbol{x},t)$. Nuclear cofilin–actin dissociates at a rate of $\gamma_{\Xi_n}$ to give $G_n(\boldsymbol{x},t)$ and $C_n(\boldsymbol{x},t)$. Finally, we assume that the nucleus contains a population of profilin, $P_n(\boldsymbol{x},t)$, which associates with $G_n(\boldsymbol{x},t)$ at a rate of $\alpha_{\Upsilon_n}$ to form nuclear profilin–actin, $\Upsilon_n(\boldsymbol{x},t)$, and dissociates from $\Upsilon_n(\boldsymbol{x},t)$ at a rate of $\gamma_{\Upsilon_n}$. The governing equations in the nucleus are therefore

$$\frac{\partial G_n}{\partial t} = D_{G_n} \nabla^2 G_n + \gamma_{\Xi_n}\Xi_n - \alpha_{\Xi_n}G_nC_n + \gamma_{\Upsilon_n}\Upsilon_n - \alpha_{\Upsilon_n}G_nP_n, \tag{13}$$

$$\frac{\partial C_n}{\partial t} = D_{C_n} \nabla^2 C_n - \alpha_{\Xi_n}G_nC_n + \gamma_{\Xi_n}\Xi_n, \tag{14}$$

$$\frac{\partial \Xi_n}{\partial t} = D_{\Xi_n} \nabla^2 \Xi_n - \gamma_{\Xi_n}\Xi_n + \alpha_{\Xi_n}G_nC_n, \tag{15}$$

$$\frac{\partial P_n}{\partial t} = D_{P_n} \nabla^2 P_n - \alpha_{\Upsilon_n}G_nP_n + \gamma_{\Upsilon_n}\Upsilon_n, \tag{16}$$

$$\frac{\partial \Upsilon_n}{\partial t} = D_{\Upsilon_n} \nabla^2 \Upsilon_n - \gamma_{\Upsilon_n}\Upsilon_n + \alpha_{\Upsilon_n}G_nP_n, \tag{17}$$

for $\boldsymbol{x} \in \Omega_n$.

At the external boundary of the cytoplasm, each species obeys the no-flux condition

$$\frac{\partial G_c}{\partial n} = \frac{\partial F_c}{\partial n} = \frac{\partial C_c}{\partial n} = \frac{\partial \Xi_c}{\partial n} = \frac{\partial P_c}{\partial n} = \frac{\partial \Upsilon_c}{\partial n} = \frac{\partial \Theta_c}{\partial n} = \frac{\partial E_c}{\partial n} = 0, \boldsymbol{x} \in \Gamma_c, \tag{18}$$

where $\boldsymbol{n}$ denotes the outwards-facing unit normal vector. At the internal boundary between the cytoplasm and the nucleus, we impose the boundary conditions

$$D_{P_c} \frac{\partial P_c}{\partial \boldsymbol{n}} = \beta_{P_n}P_n - \beta_{P_c}P_c, \tag{19}$$

$$-D_{P_n} \frac{\partial P_n}{\partial \boldsymbol{n}} = \beta_{P_c}P_c - \beta_{P_n}P_n, \tag{20}$$

$$D_{\Xi_c} \frac{\partial \Xi_c}{\partial \boldsymbol{n}} = -\beta_{\Xi_c}\Xi_c, \tag{21}$$

$$-D_{\Xi_n} \frac{\partial \Xi_n}{\partial \boldsymbol{n}} = \beta_{\Xi_c}\Xi_c, \tag{22}$$

$$D_{C_c} \frac{\partial C_c}{\partial \boldsymbol{n}} = \beta_{C_n}C_n - \beta_{C_c}C_c, \tag{23}$$

$$-D_{C_n} \frac{\partial C_n}{\partial \boldsymbol{n}} = \beta_{C_c}C_c - \beta_{C_n}C_n, \tag{24}$$

$$D_{\Upsilon_c} \frac{\partial \Upsilon_c}{\partial \boldsymbol{n}} = \beta_{\Upsilon_n}\Upsilon_n, \tag{25}$$

$$-D_{\Upsilon_n} \frac{\partial \Upsilon_n}{\partial \boldsymbol{n}} = -\beta_{\Upsilon_n}\Upsilon_n, \tag{26}$$

for $\boldsymbol{x} \in \Gamma_n$, where $\beta_{[\cdot]}$ denotes the transfer rate between the cytoplasm and the nucleus for species $[\cdot]$. The remaining nuclear boundary conditions are taken to be

$$\frac{\partial G_c}{\partial n} = \frac{\partial G_n}{\partial n} = \frac{\partial F_c}{\partial n} = \frac{\partial \Theta_c}{\partial n} = \frac{\partial E_c}{\partial n} = 0, \boldsymbol{x} \in \Gamma_n. \tag{27}$$

Simulations were performed in the Virtual Cell modelling and simulation software environment[65], using the built-in Semi-Implicit Finite Volume-Particle Hybrid (regular grid, fixed time step) solver. The domain, defined by Eqs. (1)–(4), was approximated using a rectangular Cartesian mesh comprising $201 \times 201$ grid points and all simulations used a time-step size of 0.01 time units. Default simulation parameters are as detailed in Supplementary Table 2.

As may be seen from the default simulation parameters (Supplementary Table 2), it was assumed that F-actin and the catalytic growth factor are both non-diffusing, such that $D_{F_c} = D_{E_c} = 0$. The remaining diffusion coefficients were chosen such that $D_{[\cdot]_n} \gg D_{[\cdot]_c}$ to ensure that the concentration profile of each species remains reasonably 'flat' within the nucleus (relative to the cytoplasm). We also assumed that, once introduced, the total amount catalytic growth factor is constant over time, such that $\gamma_{E_c} = 0$. In the model, the primary purpose of cofilin is to transport G-actin from the cytoplasm to the nucleus; we therefore set $\alpha_{\Xi_c} > \alpha_{\Xi_n}$ (i.e., cofilin and G-actin associate at a faster rate in the cytoplasm than in the nucleus) and $\gamma_{\Xi_c} < \gamma_{\Xi_n}$ (i.e., cofilin–actin disassociates at a faster rate in the nucleus than in the cytoplasm), so that cofilin–actin quickly dissociates upon entering the nucleus. As the profilin species is used to transport G-actin from the nucleus to the cytoplasm, we set $\alpha_{\Upsilon_n} > \alpha_{\Upsilon_c}$ (i.e., profilin and G-actin associate at a faster rate in the nucleus than in the cytoplasm) and $\gamma_{\Upsilon_n} < \gamma_{\Upsilon_c}$ (i.e., profilin–actin disassociates at a faster rate in the cytoplasm than in the nucleus), so that profilin–actin quickly dissociates upon entering the cytoplasm. To ensure that the equilibrium concentration of cofilin is approximately the same on either side of the nuclear boundary, we set $\beta_{C_c} = \beta_{C_n}$; similarly, we set $\beta_{P_c} = \beta_{P_n}$ so that the equilibrium concentration of profilin is approximately the same on either side of the nuclear boundary. Wet lab experiments have revealed that, in the absence of a catalytic growth factor, the concentration of G-actin in the nucleus is typically higher than the concentration of G-actin in the cytoplasm; given sufficiently large values for $\gamma_{\Xi_n}$ and $\gamma_{\Upsilon_c}$, this implies that $\beta_{\Xi_c} > \beta_{\Upsilon_n}$. Finally, the first catalytic rate parameter, $\mu_1$, was chosen to ensure that G-actin is polymerized at a faster rate in the presence of a catalytic growth factor (as observed experimentally), whilst the second catalytic rate parameter, $\mu_2$, was chosen to ensure that cofilin is rapidly phosphorylated in the presence of a catalytic growth factor (which should have the effect of reducing the amount of cofilin that is available to transport G-actin from the cytoplasm to the nucleus).

To determine appropriate initial conditions, the model system was simulated using the default parameters outlined above (with $E_c(\boldsymbol{x},t) = 0$, $\forall \boldsymbol{x} \in \Omega_c$) until it had approached equilibrium, and then the average concentration of each species (across its associated domain) was used as a spatially constant initial value in subsequent simulations. The computed initial value for each species are detailed in Supplementary Table 3. It is noteworthy that the equilibrium concentration profile for each species is not necessarily flat, so our simulations (which use a spatially constant value as the initial condition for each species) do not start at steady state.

Finally, the initial condition for the catalytic growth factor was chosen to be

$$E(\boldsymbol{x}, 0) = \begin{cases} E_{c_{init}}, & \|\boldsymbol{x} - \boldsymbol{p}_3\| < r_3, x \in \Omega_c \\ 0, & \text{otherwise} \end{cases}, \qquad (28)$$

**Reporting summary**. Further information on research design is available in the Nature Research Reporting Summary linked to this article.

## Data availability

All raw data were generated at the CRUK Beatson Institute. All raw data and data derived from this, which are relevant to this study, are available from the corresponding author [J.C.N.] on request. The SILAC Nuclear Capture Proteome data generated in this study have been deposited in the ProteomeXchange Consortium via the PRIDE partner repository under the accession code PXD027268. The RNA sequencing data generated in this study have been deposited in the GEO – NCBI – NIH repository under the accession number GSE179901. The EphA2-TurboID proteomic data generated in this study have been deposited in the ProteomeXchange Consortium via the PRIDE partner repository under the accession code PXD027217. The remaining data are available within the Article and Supplementary Information. Source data are provided with this paper.

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

## Acknowledgements

This work was funded by Cancer Research UK (Core grants: A17196 and A31287, JCN grant: A18277 and SZ grant: A29800). Many thanks to Drs. Christian Baarlink and Robert Grosse for the generous gift of H1299 cells expressing GFP-MRTF. Many thanks to Professor Anne Ridley for the generous gift of GFP-tagged RhoG expression vector, and to Dave Bryant and Konstantina Nikolatou for provision of the TurboID construct and for invaluable assistance in making the TurboID-EphA2s.

## Author contributions

Experimental work and primary data analysis were performed by S.M., M.M., A.P.-G., L.M., G.R.B. and J.C.N. Computational modelling was designed and performed by M.N. Bioinformatic analyses were performed by H.H. and A.H. S.Z. and S.L. designed and supervised the proteomic approach to characterizing the plasma membrane to nuclear traffickome. Experiments were conceived and designed by S.M., M.N., S.Z. and J.C.N. S.M. and J.C.N. wrote the manuscript.

## Competing interests

The authors declare no competing interests.
