## [Peer Review File · Nature Communications]

REVIEWER COMMENTS:

Reviewer #1 (Remarks to the Author):

As stated in the original critique, the hypothetic model proposed by the authors is appealing, and their findings are interesting and transformative. Unfortunately, new experiments in the revised manuscript somewhat lessened my enthusiasm. In particular, new electron microscopy (EM) data are not convincing. First, it is unclear why EM was performed with the nuclear fraction, which introduces additional variabilities in the preservation of an intact distribution of compartments and uncertainties in interpretations, rather than with intact cells. Second, there is only a single gold particle marking a biotinylated protein of an unknown identity, and it is unclear whether this label is on an intraluminal vesicle or on the cytoplasmic side of the limiting membrane of MVB. The latter localization would be impossible because the label must be on an extracellular part of a membrane protein and thus oriented into the lumen of an MVB. Finally, there is no evidence for the presence of a nuclear pore in the vicinity of docked MVB.

With regards to other concerns, the geometry of EphA2 interaction through their compact juxtamembrane regions with importins/nuclear pore complex remains puzzling, and the mechanisms by which Met promotes internalization of EphA2 remain unknown.

Clearly, localization of RhoG in the pericentriolar/Golgi area, where a bulk of endosomes is typically concentrated, generates an increased probability of the receptor-RhoG interaction, although such interaction would be via a “trans-compartment” mechanism: RhoG is associated with the membrane compartments constitutively located in the Golgi area, where newly formed (after Met activation) EphA2 endosomes are arriving. Overall, a new modified model involving actin polymerization in this juxta-Golgi area is logical and supported by the data. However, the necessity of a direct EphA2 docking to nuclear pores for RhoG activation is difficult to explain theoretically and reconcile with the microscopy data. For example, if NLS mutants of EphA2 are efficiently transported to the juxta-Golgi area, they should have the same probability of the interaction with RhoG as the wild-type EphA.

Reviewer #2 (Remarks to the Author):

This is a thoroughly revised manuscript. The authors have done an excellent job in addressing all my previous concerns. This will be a nice contribution to the field of RTK (Eph2A) signaling and nuclear biology.

Reviewer #4 (Remarks to the Author):

The authors have generally responded well to my comments. I was positive about the previous version and with a few additional minor changes would support publication of this work.

The authors provide new CHIP data about MRTF occupancy which answers the issue we raised. However, the data are not referred to correctly in the text ie the wrong part of the figure is mentioned.

Also on this point, it is possible that the reduced MRTF signal is due to reduced SRF expression rather than a direct effect on MRTF (as SRF recruits MRTF). AS SRF expression declines, can the authors design experiments to distinguish these possibilities (eg constitutively express SRF or show that rescuing SRF levels does not rescue the MRTF binding defect) or at least point out this complication in the text.

The authors have made efforts to explain things more in the text but there is still no explanation about how DNAase1 staining reveals anything about G actin levels (and I have no idea about the connection myself). A brief explanation of the LifeACT system would also be useful.

RESPONSE TO REVIEWERS

Reviewer #1 – comment 1

“First, it is unclear why EM was performed with the nuclear fraction, which introduces additional variabilities in the preservation of an intact distribution of compartments and uncertainties in interpretations, rather than with intact cells. Second, there is only a single gold particle marking a biotinylated protein of an unknown identity, and it is unclear whether this label is on an intraluminal vesicle or on the cytoplasmic side of the limiting membrane of MVB. The latter localization would be impossible because the label must be on an extracellular part of a membrane protein and thus oriented into the lumen of an MVB. Finally, there is no evidence for the presence of a nuclear pore in the vicinity of docked MVB.”

Response

We agree with this reviewer that the way in which the EM was performed in the previous submission of the paper did not provide evidence that EphA2-positive endosomes interact with the nuclear pore. We have, therefore, removed these data from the paper.

To explore how EphA2 might interact with the nuclear membrane, we deployed a proximity-labelling approach. We generated constructs in which either wild-type or NLS-mutated EphA2s were fused to an engineered biotin ligase, TurboID, which catalyses addition of biotin to proteins within its close proximity (<10nm). These biotinylated proteins were then recovered using streptavidin beads and analysed by label-free mass spectrometry-based proteomics. This analysis indicated that the proximity proteome of EphA2 comprised several nuclear pore complex and nuclear pore-related proteins, and that interaction of these with EphA2 was favoured when its NLS was intact. We validated a selection of the most prominent nuclear pore hits by Western blotting, and this demonstrated that EphA2 interacted with proteins located at the outer and central pore and cytoplasmic ring of the nuclear pore complex, but not with underlying inner nuclear membrane proteins, such as laminA/C. These new data are presented in Figure 3b – e, Supplementary Fig. 1b – d, and Supplementary spreadsheet 2.

Reviewer #1 – comment 2

“With regards to other concerns, the geometry of EphA2 interaction through their compact juxtamembrane regions with importins/nuclear pore complex remains puzzling.....”

Response

In addition to providing *de facto* evidence for an NLS-dependent interaction of EphA2 with the nuclear pore, we consider this approach appropriate to address this reviewer’s previous

concerns on the topography of endosome-nucleus interaction. Interestingly, our proximal proteomic analysis (and subsequent Western blotting validation) revealed RanBP2 to be a proximal interactor of EphA2. RanBP2 (also known as Nup358) is a high molecular weight component of the filaments present on the cytoplasmic side of the nuclear pore complex. We have also identified and validated other components of the cytoplasmic filament array, such as Nup214 and Nup88 as EphA2 interactors. RanBP2 has been previously identified as a potential docking site for importin- β and other importins that facilitate nuclear import of various cargoes^{1, 2}. Since the cytoplasmic filaments are relatively flexible, and are also known to participate in the docking of macromolecular structures such as viral capsids³⁻⁵, it seems plausible to envisage that the interaction between RanBP2, nuclear importins and EphA2's cytodomain would mediate association of EphA2-positive endosomes with the nuclear membrane.

Reviewer #1 – comment 3

“.....the mechanisms by which Met promotes internalization of EphA2 remain unknown”

Response

Our previous data showed that Rab17 was the main Rab GTPase contributing to the endocytosis and trafficking of EphA2 to the nuclear vicinity. We observed that Rab17 colocalised with EphA2 in vesicles, and that these vesicles had the ability to physically attach to the nuclear surface. Moreover, we showed that HGF-dependent internalisation of EphA2 is dependent on Rab17 expression and that Rab17 is necessary to induce the expression of SRF/MRTF-target genes when the cells are stimulated with HGF.

We have now added further data to clarify the pathway responsible for endocytosis of EphA2. We show that an siRNA that suppresses expression of the clathrin heavy chain, and which we show to block endocytosis of the transferrin receptor (a canonical cargo of clathrin-dependent endocytosis), significantly inhibits both basal and HGF-driven EphA2 internalisation. Consistently, clathrin knockdown significantly opposes the HGF-driven transcription of SRF/MRTF-target genes that we have established to be EphA2 and nuclear-capture dependent. These new data are presented in Fig. 2a-b and Supplementary Fig. 3a.

Reviewer #1 – comment 4

Clearly, localization of RhoG in the pericentriolar/Golgi area, where a bulk of endosomes is typically concentrated, generates an increased probability of the receptor-RhoG interaction, although such interaction would be via a “trans-

compartment” mechanism: RhoG is associated with the membrane compartments constitutively located in the Golgi area, where newly formed (after Met activation) EphA2 endosomes are arriving. Overall, a new modified model involving actin polymerization in this juxta-Golgi area is logical and supported by the data. However, the necessity of a direct EphA2 docking to nuclear pores for RhoG activation is difficult to explain theoretically and reconcile with the microscopy data. For example, if NLS mutants of EphA2 are efficiently transported to the juxta-Golgi area, they should have the same probability of the interaction with RhoG as the wild-type EphA.

Response

We agree with this analysis. However, the NLS mutants of EphA2 are less efficiently transported to the juxta-Golgi area than their wild-type counterparts (see Fig. 6a; Supplementary Fig. 1b)

Reviewer #4 – comment 1

“However, the data are not referred to correctly in the text ie the wrong part of the figure is mentioned.”

Response

We have addressed this issue by re-structuring the results section, including the text describing the ChIP experiments.

Reviewer #4 – comment 2

“Also on this point, it is possible that the reduced MRTF signal is due to reduced SRF expression rather than a direct effect on MRTF (as SRF recruits MRTF). AS SRF expression declines, can the authors design experiments to distinguish these possibilities (eg constitutively express SRF or show that rescuing SRF levels does not rescue the MRTF binding defect) or at least point out this complication in the text.”

Response

To address this point we have now included the following statement in the appropriate results sub-section on page 8: *“Although this clearly demonstrates a requirement for EphA2 signalling in HGF-driven recruitment of MRTF to its target genes, it remains unclear how EphA2 knockdown might impact on MRTF/SRF protein levels under basal conditions.”*

Reviewer #4 – comment 3

“The authors have made efforts to explain things more in the text but there is still no explanation about how DNAase1 staining reveals anything about G actin levels (and I have no idea about the connection myself). A brief explanation of the LifeACt system would also be useful.”

Response

The first report of actin as an interactor of DNaseI was published in 1974 by Elias Lazarides and Uno Lindberg ⁶, where they describe how the binding of actin to the enzyme inhibits its nuclease activity. Subsequently, DNaseI has been included in the list of known actin binding proteins. The binding to actin is mediated through domains II and IV, and despite that DNaseI can bind to both monomeric and polymerised actin, it does so with widely different affinities. The binding of DNaseI to monomeric actin occurs with a Kd of 1nM whereas the binding to the actin filament requires has a Kd of 100µM ⁶. Therefore, by taking advantage of this difference, using fluorophore-labelled DNaseI at low concentrations, G-actin can be specifically stained. A brief description has been added to the appropriate results text on page 9.

Lifeact is a 17-amino-acid peptide sequence extracted from the yeast protein Abp140, which is exclusively in lower eukaryotes but not mammalian cells ⁷. This 17 amino acid stretch is sufficient to bind specifically to F-actin and it does so without influencing actin filament stability. Therefore, when fused to a fluorescent protein such as GFP or mCherry, Lifeact can be used to label F-actin structures and to monitor their dynamics *in vivo*.

References

1. Hamada, M. *et al.* Ran-dependent docking of importin-beta to RanBP2/Nup358 filaments is essential for protein import and cell viability. *J Cell Biol* **194**, 597-612 (2011).
2. Walde, S. *et al.* The nucleoporin Nup358/RanBP2 promotes nuclear import in a cargo- and transport receptor-specific manner. *Traffic* **13**, 218-233 (2012).
3. Brass, A.L. *et al.* Identification of host proteins required for HIV infection through a functional genomic screen. *Science* **319**, 921-926 (2008).
4. Di Nunzio, F. *et al.* Human nucleoporins promote HIV-1 docking at the nuclear pore, nuclear import and integration. *PLoS One* **7**, e46037 (2012).
5. Schaller, T. *et al.* HIV-1 capsid-cyclophilin interactions determine nuclear import pathway, integration targeting and replication efficiency. *PLoS Pathog* **7**, e1002439 (2011).
6. Lazarides, E. & Lindberg, U. Actin is the naturally occurring inhibitor of deoxyribonuclease I. *Proc Natl Acad Sci U S A* **71**, 4742-4746 (1974).
7. Riedl, J. *et al.* Lifeact: a versatile marker to visualize F-actin. *Nat Methods* **5**, 605-607 (2008).

REVIEWERS' COMMENTS

Reviewer #5 (Remarks to the Author):

As requested by the Editor, I focused my review on the technical aspects of MS-based proteomics. The following points should be addressed prior to consideration for publication in Nat. Commun:

(1) Figure 1b and page 5 line 127: The fold-change of EPHA2 is not so large and the reproducibility is low (1.14-fold for Exp1 and 1.85-fold for Exp2). Looking at this data, I think it is quite an exaggeration to say that "EphA2 was prominent amongst these".

(2) Figure 3c: Why is the x-axis of this volcano plot plotted with the Student's t-test difference instead of the more common Log2 (fold-change (WT/NLS1))? What is the definition of the Student's t-test difference (how was it calculated)?

(3) Figure 3c: Threshold for determining significantly enriched proteins should be clarified. Typically, it is a fold-change of at least 1.5 and a p-value of less than 0.05. Do the identified nucleoproteins meet the criteria?

Reviewer #6 (Remarks to the Author):

This manuscript reports a complex series of events by which activation of the MET receptor tyrosine kinase (RTK) leads to scattering and invasion behaviors of cancer cells. These events include (i) MET-induced, Rab17- and clathrin-dependent endocytosis of another RTK, EphA2, (ii) transport of EphA2-positive endosomes toward the juxtannuclear area, (iii) capture of these endosomes on the outer surface of the nucleus through interaction between the nuclear import machinery and a nuclear localization sequence in EphA2, (iv) RhoG-dependent phosphorylation of cofilin to oppose nuclear import of G-actin, (v) transcription of myocardin-related transcription factor (MRTF)/serum-response factor (SRF)-target genes. Because of the expansive nature of this work, each step is not analyzed in much detail. Rather, the value of this study lies in the connection of multiple, distinct cellular processes. The use of NLS mutants as controls in many of the experiments is a definite strength. I trust that the evidence on individual processes has been evaluated by subject-matter experts, so I will limit my opinion to the responses to the reviewers' latest comments.

Reviewer #1 – comment 1

Although I haven't seen the EM data in the previous version of the paper, I agree with reviewer #1 that an image of an MVB with a single gold particle representing a surface-biotinylated protein next to the nucleus, without a nuclear pore in sight, is not convincing. I acknowledge that such evidence would be difficult to obtain using the authors' method. I wish the authors had tried correlative light electron microscopy (CLEM), proximity ligation assays (PLA), or FRET to demonstrate the presumed interaction of EphA2 with nuclear pore proteins. However, I must say that the new proximity biotinylation data added by the authors make a good case for an NLS-dependent proximity of EphA2 to the nuclear pore.

Reviewer #1 – comment 2

I wondered the same thing: is the putative NLS in EphA2 accessible? I would have liked to see direct biochemical evidence for the interaction of EphA2 through its putative NLS with importin. The evidence presented in the paper is rather indirect, although the processes reported seem to be dependent on the NLS, and that is a good control.

Reviewer #1 – comment 3

The demonstration that clathrin is required for EphA2 internalization and HGF-driven transcription of SRF/MRTF-target genes doesn't really clarify the mechanism by which MET activation induces Eph2A internalization. The question remains as to what MET activation really does. Does it phosphorylate EphA2 or some component of the endocytic machinery? How does this phosphorylation cause EphA2 internalization? I think this is what Reviewer #1 – and I – would like to see.

Reviewer #1 – comment 4

I don't understand the authors' finding/response that "NLS mutants of EphA2 are less efficiently transported to the juxta-Golgi area than their wild-type counterparts." How does an NLS mediate translocation to the juxta-nuclear area? I would understand that it mediates docking to the nuclear pore, but not the centripetal transport itself. This should be clarified in the final version of the manuscript.

Reviewer #4 – comment 1

The authors made the necessary corrections.

Reviewer #4 – comment 2

I am satisfied with the authors' response.

Reviewer #4 – comment 3

I am satisfied with the authors response.

Reviewer #5 – comment 1

(1) Figure 1b and page 5 line 127: The fold-change of EPHA2 is not so large and the reproducibility is low (1.14-fold for Exp1 and 1.85-fold for Exp2). Looking at this data, I think it is quite an exaggeration to say that “EphA2 was prominent amongst these”.

Response

We have now amended the text and replaced it with: “The results obtained indicated that activation of MET by HGF promoted translocation of a number of proteins, mainly receptors, from the plasma membrane to the nucleus, and another RTK, EphA2 was amongst these..”. See page 5, line 22.

Reviewer #5 – comment 2

(2) Figure 3c: Why is the x-axis of this volcano plot plotted with the Student's t-test difference instead of the more common Log2 (fold-change (WT/NLS1))? What is the definition of the Student's t-test difference (how was it calculated)?

Response

We have used Perseus software to analyse proteomic data and determine the EphA2 proximity interactome, and this entails the derivation and use of Student's t-test differences. We have now provided a more comprehensive description of this approach in the ‘Proteomics data analysis’ subsection of the ‘Proteomics for determination of the EphA2 proximity interactome’ methods section. Indeed, on page 24, line 21-26, we have appended the following description:

Data presented in the volcano plots were expressed by using the “Student's t-test difference” metric which corresponds, for a given protein, to the average of the log₂-transformed LFQ intensities measured in each replicate from condition 1 (WT), subtracted from the counterpart average in condition 2 (either NLS1 or NLS2): e.g. [average log₂ transformed LFQ intensity in condition A] – [average log₂ transformed LFQ intensity in condition B].

An alternative approach, which is suggested by the reviewer, is to calculate the “Fold change” for a given protein, by dividing the average LFQ intensities measured for a protein in each replicate of condition1 (WT) by the counterpart in condition2 (NLS1). The result may then be log₂-transformed to obtain the “log₂ Fold Change”.

Both approaches are valid. However, calculation of “Student's t-test difference” is the approach that is customarily used in conjunction with the Perseus software that we deploy for this study.

Reviewer #5 – comment 3

(3) Figure 3c: Threshold for determining significantly enriched proteins should be clarified. Typically, it is a fold-change of at least 1.5 and a p-value of less than 0.05. Do the identified nucleoproteins meet the criteria?

Response

We agree with the reviewer that the criteria typically used for determining significantly-enriched proteins are to consider those displaying a fold increase of >1.5 and a p value <0.05. As we have used Student's t-test difference, rather than fold-change, for our analysis, we are not able to directly apply these criteria. Nevertheless, hits that display a Student's t-test value (denoted on the y-axis in Fig. 3c) of greater than $-\text{Log} = 1.4$ are considered to be statistically significant, as this corresponds to a p-value of <0.05. Indeed, using this criterion the majority of nuclear pore components that we have highlighted in Fig. 3c display statistically significantly enhanced association with EphA2^{WT} over the EphA2^{NLS1} or EphA2^{NLS2} mutants. We have appended the following text to the legend to Fig. 3c (page 39, line 27):

The significance scores ($-\text{Log}$ Student's t-test p-value, n=5 independent experiments) are plotted on the y-axes. Student's t-test values greater than $-\text{Log} 1.4$ (i.e., less than $p = 0.05$) are statistically significant. The proteins highlighted and annotated correspond to all the nuclear pore components that were identified in the proximity interactome regardless of their enrichment.

Reviewer #6 – comment 2

I wondered the same thing: is the putative NLS in EphA2 accessible? I would have liked to see direct biochemical evidence for the interaction of EphA2 through its putative NLS with importin. The evidence presented in the paper is rather indirect, although the processes reported seem to be dependent on the NLS, and that is a good control.

Response

To discuss this point, we have now appended the following sentence to the discussion on page 14, line 29:

Thus, although we have not yet demonstrated direct association between EphA2's cytotail and an importin, we submit that NLS-mediated recruitment of EphA2-positive endosomes to the nuclear membrane to allow signalosome assembly should be added to the list of tasks performed by elements of the nuclear import machinery.

Reviewer #6 – comment 3

The demonstration that clathrin is required for EphA2 internalization and HGF-driven transcription of SRF/MRTF-target genes doesn't really clarify the mechanism by which MET activation induces Eph2A internalization. The question remains as to what MET activation really does. Does it phosphorylate EphA2 or some component of the endocytic machinery? How does this phosphorylation cause EphA2 internalization? I think this is what Reviewer #1 – and I – would like to see.

Response

We have previously shown that HGF drives EphA2 internalisation by promoting Akt-mediated phosphorylation of Ser⁸⁹⁷ in its cytotail. Thus, we now know that HGF-driven endocytosis of EphA2 is driven by this phosphorylation event, by Rab17 and that this process is clathrin-dependent. We have re-structured the text of the appropriate results section to illustrate this – see page 6 , line 1:

HGF is established to promote phosphorylation of the Ser⁸⁹⁷ in EphA2's cytotail^{1, 2}, and we have previously shown that this event is required for HGF-driven internalisation of EphA2³. To further elucidate factors required for HGF-driven packaging of EphA2 into endosomes, we deployed an siRNA to oppose clathrin heavy chain expression (Fig. 2a), and which blocks endocytosis of the transferrin receptor (the hallmark cargo of clathrin-coated pits) (Fig. 2b). This indicated that EphA2 internalisation (both in the absence and presence of HGF) is strongly clathrin-dependent (Fig. 2c). We next tested the involvement of a battery of Rab GTPases and other endosomal regulators known to be involved in cell adhesion and migration (including Rabs 4, 5, 7, 11, 17, 21 and 25, CD63 and LAMP1/2). This highlighted Rab17 as being required for both HGF-driven internalisation (Fig. 2d) and nuclear-capture (Fig. 2e) of EphA2.

Reviewer #6 – comment 4

I don't understand the authors' finding/response that "NLS mutants of EphA2 are less efficiently transported to the juxta-Golgi area than their wild-type counterparts." How does an NLS mediate translocation to the juxta-nuclear area? I would understand that it mediates docking to the nuclear pore, but not the centripetal transport itself. This should be clarified in the final version of the manuscript.

Response

We included this comment in the rebuttal to reviewer #1, but not in the manuscript. The comment was provided to discuss reviewer #1's concern that nuclear-capture should not be necessary to move receptors into the general proximity of RhoG in the juxta-Golgi region. Presumably, reviewer #1 is asserting that there are other ways of moving receptors to the

juxta-Golgi region without having to invoke nuclear-capture. We agree with this. However, it is clear from looking at images in Fig. 6a (including its associated movies 4 & 5) and Supplementary figure 1b that EphA2s with mutated NLSs are not only less associated with the nuclear surface, but also found to be less abundant in the juxta-nuclear/Golgi area in general. It is possible, therefore, that NLSs and nuclear-capture contribute to concentration of EphA2-positive endosomes in the general vicinity of the nucleus, as well as specifically to their physical capture on the nuclear surface. Nevertheless, despite clear observations, made in Fig. 6a and Supplementary figure 1b, that EphA2 NLS mutants are less abundant in the juxta-Golgi/nuclear region, the arguments that we make above as to how this might occur are rather speculative and are, as such, best not included in the published text.

References

1. Miao, H. *et al.* EphA2 mediates ligand-dependent inhibition and ligand-independent promotion of cell migration and invasion via a reciprocal regulatory loop with Akt. *Cancer Cell* **16**, 9-20 (2009).
2. Zhou, Y. *et al.* Crucial roles of RSK in cell motility by catalysing serine phosphorylation of EphA2. *Nat Commun* **6**, 7679 (2015).
3. Gundry, C. *et al.* Phosphorylation of Rab-coupling protein by LMTK3 controls Rab14-dependent EphA2 trafficking to promote cell:cell repulsion. *Nat Commun* **8**, 14646 (2017).